# Pla2g12b drives expansion of triglyceride-rich lipoproteins

James H. Thierer[1,2], Ombretta Foresti ®[3], Pradeep Kumar Yadav[4,5], Meredith H. Wilson[1,2], Tabea O. C. Moll ®[1,2], Meng-Chieh Shen[1], Elisabeth M. Busch-Nentwich[6], Margaret Morash[7], Karen L. Mohlke ®[8], John F. Rawls ®[7], Vivek Malhotra ®[3,9,10], M. Mahmood Hussain ®[4] & Steven A. Farber ®[1,2] ✉

Vertebrates transport hydrophobic triglycerides through the circulatory system by packaging them within amphipathic particles called Triglyceride-Rich Lipoproteins. Yet, it remains largely unknown how triglycerides are loaded onto these particles. Mutations in Phospholipase A2 group 12B (*PLA2G12B*) are known to disrupt lipoprotein homeostasis, but its mechanistic role in this process remains unclear. Here we report that PLA2G12B channels lipids within the lumen of the endoplasmic reticulum into nascent lipoproteins. This activity promotes efficient lipid secretion while preventing excess accumulation of intracellular lipids. We characterize the functional domains, subcellular localization, and interacting partners of PLA2G12B, demonstrating that PLA2G12B is calcium-dependent and tightly associated with the membrane of the endoplasmic reticulum. We also detect profound resistance to atherosclerosis in PLA2G12B mutant mice, suggesting an evolutionary tradeoff between triglyceride transport and cardiovascular disease risk. Here we identify PLA2G12B as a key driver of triglyceride incorporation into vertebrate lipoproteins.

Lipids are hydrophobic molecules that play essential roles in membrane synthesis, cellular signaling, and energy storage. Multicellular organisms transport lipids between tissues by packaging them into micelle-like particles called lipoproteins[1]. Microsomal triglyceride transfer protein (MTP)[2] mediates lipoprotein production by transferring lipids from the endoplasmic reticulum (ER) membrane to a carrier protein called Apolipoprotein-B (APOB)[3,4]. APOB is translated by ER-bound ribosomes and passes through the translocon[5]. Within the ER, the hydrophobic portions of APOB are stabilized through interactions with the lumenal face of the ER membrane as well as the co-translational addition of lipids by MTP[6]. If insufficient lipids are available or MTP activity is compromised, APOB is rapidly degraded via the ER-associated degradation (ERAD) pathway[7].

Vertebrate MTP transfers a variety of lipids to APOB, including triglycerides, phospholipids, and cholesteryl-esters[8,9]. As their name suggests, Triglyceride-Rich Lipoproteins (TRLs) are predominantly composed of triglycerides, which account for >60% of the particle mass. TRLs secreted from the liver are referred to as very-low-density lipoproteins (VLDL), whereas those secreted from the intestine are called chylomicrons (CM, Fig. 1A). Intestine-derived chylomicrons are significantly larger than VLDL and carry a truncated form of APOB called APOB-48 in humans, but there is no evidence of this truncated

[1]Department of Embryology, Carnegie Institution for Science, Baltimore, MD 21218, USA. [2]Johns Hopkins University in Baltimore, Maryland Department of biology, Baltimore, MD 21218, USA. [3]Centre for Genomic Regulation (CRG), The Barcelona Institute for Science and Technology, Barcelona 08003 ES, Spain. [4]Department of Foundations of Medicine, NYU Long Island School of Medicine, Mineola, NY 11501, USA. [5]Department of Botany, Faculty of Science, University of Allahabad, Prayagraj, India. [6]School of Biological and Behavioural Sciences, Queen Mary University of London, London E1 4NS GB, UK. [7]Department of Molecular Genetics and Microbiology, Duke University, Durham, NC 27708, USA. [8]Department of Genetics, University of North Carolina, Chapel Hill, NC 27599, USA. [9]Universitat Pompeu Fabra, Barcelona, Spain. [10]Institució Catalana de Recerca i Estudis Avançats, Barcelona, Spain. ✉e-mail: sfarber3@jhu.edu

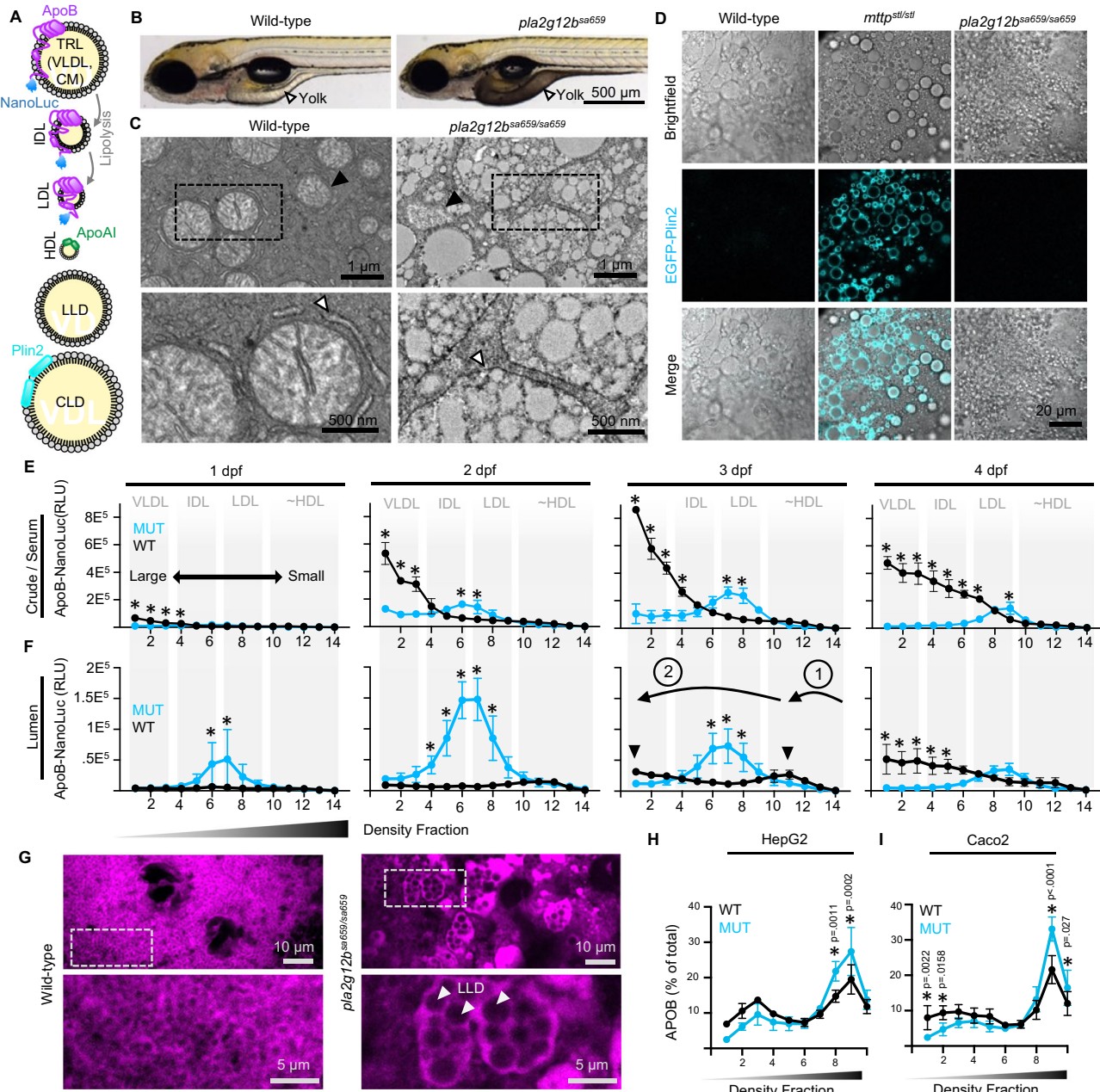

**Fig. 1 | Pla2g12b directs ER triglycerides to nascent TRLs. A** Schematic representation of lipid micelle types and their markers. Elements of this panel have been adapted from previous work[10] under Creative Commons Attribution 4.0 International License. **B** Representative images of larval zebrafish at 4 days post fertilization (dpf) illustrating darkening of the normally translucent yolk (white arrowhead) in *pla2g12b*[sa659] mutants. Scale bar = 500 μm, *n* = 9. **C** Electron micrographs of the Yolk Syncytial Layer (YSL) and enlarged regions of interest (dashed boxes) show that the presumptive ER lumen (white arrowhead) is distended with lipid micelles in *pla2g12b*[sa659] mutants. Circular features in wild-type micrographs are mitochondria (black arrowhead). Scale bar = 1 μm in upper panels, 500 nm in lower panels. Image is representative of *n* = 3 biological replicates. **D** Confocal images of the EGFP-Plin2 reporter of CLDs. CLDs are undetectable in *pla2g12b*[sa659] mutants despite significant intracellular lipid accumulation. Scale bar = 20 μm, *n* ≥ 9. **E** Density gradient profiles of ApoB-NanoLuc (relative luminescence units, RLU) in crude homogenate illustrating production of significantly more dense TRLs by *pla2g12b*[sa659] mutants (cyan trace). Data are mean ± s.d. of *n* = 3 biological replicates. Two-way ANOVA

*p* < 0.005, asterisks denote *p* < 0.05 by Šídák's multiple comparison test. **F** Density gradient profiles of ApoB-NanoLuc in ER-lumenal (microsomal) extracts from 1–4 dpf. Wild-type extracts exhibit a bimodal density distribution (black arrowheads), reflective of a two-step lipid transfer process (denoted as "1" and "2" with black arrows). Data are mean ± s.d. of *n* = 3 biological replicates. Two-way ANOVA *p* < 0.05, an asterisk denotes *p* < 0.05 by Šídák's multiple comparison test. **G** Confocal images of the fluorescent ER-marker ER-tdTomato and enlarged regions of interest (dashed boxes) showing lipid inclusions (white arrowheads) within the ER lumen of *pla2g12b*[sa659] mutants. Images are representative of *n* ≥ 4 samples per group. Scale bar = 10 μm in upper panels, 5 μm in lower panels. **H** Density gradient profiles of secreted TRLs in the media of cultured human HepG2 and **I** Caco2 cells show that *PLA2G12B*[−/−] mutants also secrete abnormally dense TRLs. Data are mean ± s.d. of *n* ≥ 4 biological replicates. Two-way ANOVA *p* < 0.0001, asterisk denotes *p* < 0.05 by Šídák's multiple comparison test. Source data are provided as a Source Data file.

form being produced in zebrafish[10]. Secreted TRLs are digested or lipolyzed by vascular lipases, giving rise to smaller triglyceride-poor species such as intermediate and low-density lipoproteins (IDL and LDL, Fig. 1A) which are subsequently taken up through receptor-mediated endocytosis. By contrast, invertebrate MTP can only transfer phospholipids and cholesteryl-esters, but not triglycerides, to nascent lipoproteins[11–13]. As a result, invertebrates do not secrete TRLs, but instead secrete small triglyceride-poor lipoproteins. The acquisition of triglyceride transfer activity by vertebrate MTP was therefore an essential step in the evolution of TRLs in the vertebrate lineage.

In addition to transferring lipids to APOB, MTP can also create APOB-free lipid micelles within the ER lumen called lumenal lipid droplets (LLDs, Fig. 1A)[14,15]. The biological function of LLDs remains ill-defined. It has been proposed that LLDs may fuse with nascent TRLs[16,17], delivering a large bolus of lipids to nascent lipoproteins prior to their secretion. However, these fusion events have never been directly observed[18,19]. It is therefore unclear whether LLDs are an essential component of the TRL biogenesis pathway or an unintended by-product of lumenal lipid transfer activity.

TRL expansion appears to occur in two phases[17,20,21]. The first phase involves the direct transfer of a small amount of lipids to APOB by MTP, forming a small nascent particle with densities similar to high-density lipoproteins (HDL). Subsequently, a second phase of lipidation occurs whereby a large amount of lipids (predominantly triglycerides) are rapidly added to the particle core, creating a mature TRL. This second phase of lipidation is not blocked by inhibitors of MTP lipid transfer activity[17], suggesting it proceeds through a distinct mechanism from the first phase[4]. As mentioned above, it has been speculated that fusion between small TRLs and LLDs could explain the rapid expansion of TRLs in this second phase of lipidation, but fusion-independent models have not been conclusively ruled out.

Triglycerides are energy-dense metabolic fuels, carrying more than twice the energy per gram as carbohydrates and proteins[22]. By facilitating solubilization of hydrophobic triglycerides in the aqueous circulatory system, the evolution of TRLs enabled vertebrates to transport metabolic energy between tissues with unprecedented efficiency. However, introducing TRLs into the bloodstream can also result in ectopic accumulation of lipids in the arterial wall and lead to formation of atherosclerotic plaques[23,24]. Atherosclerotic cardiovascular disease persists as one of the leading causes of death worldwide[25], creating an urgent unmet need to develop effective strategies to reduce triglyceride levels in the bloodstream.

Pharmacological inhibitors of MTP drastically reduce serum TRL levels and cardiovascular disease risk[26], but cause several intolerable side effects resulting from severe disruption of lipid secretion via the lipoprotein pathway[27]. It has been proposed that selectively blocking the transfer of triglycerides to nascent TRLs would offer protection from atherosclerosis while eliminating unwanted side effects by allowing phospholipid and cholesterol transport to continue unperturbed[12]. However, this therapeutic strategy has not been tested experimentally, nor have any chemical inhibitors been identified that selectively inhibit the triglyceride transfer activity of MTP.

It also remains unclear whether the triglyceride transfer activity of vertebrate MTP is sufficient to drive TRL expansion, or whether additional cofactors are required. Numerous genes have been linked to vertebrate TRL biogenesis[28–31], yet the precise mechanism by which vertebrates generate TRLs remains largely speculative. Understanding the mechanistic basis of TRL expansion would significantly advance our understanding of triglyceride metabolism and highlight potential therapeutic targets to prevent hyperlipidemia and atherosclerosis.

Here we perform a detailed investigation of Phospholipase A2 group 12B (PLA2G12B), one of a growing list of genes implicated in TRL homeostasis but lacking mechanistic insight. We find that despite being a member of the secreted phospholipase gene family, PLA2G12B has evolved a role in TRL expansion. We discover a variety of functional motifs within PLA2G12B that promote its retention within the ER, including a signal peptide, an ER-retention motif, two hydrophobic motifs that may anchor it within the membrane, and a calcium-binding domain[32]. We conclude that PLA2G12B associates with the lumenal face of the ER membrane where it enhances the transfer of triglycerides into nascent APOB-containing lipoproteins, thus enabling vertebrate animals to produce TRLs.

## Results
### Partitioning of ER Lipids
MTP is involved in the formation of two types of micelles found in the ER lumen: ApoB-containing and ApoB-free[19]. Genetic or pharmacological disruption of Mtp in zebrafish leads to accumulation of cytoplasmic lipid droplets (CLDs) in the yolk syncytial layer (YSL)[12], the embryonic organ responsible for packaging maternally-deposited yolk lipids into TRLs. Lipid retention changes the optical properties of the normally translucent zebrafish YSL, causing it to become opaque and appear dark when viewed with transmitted light[12]. We reasoned that this readily visible darkened-yolk phenotype presents a powerful system to identify additional genes responsible for TRL expansion.

We identified an existing mutant (sa659) in the zebrafish mutation project collection[33] that exhibits the dark-yolk phenotype by 4 days post fertilization (dpf, Fig. 1B). This allele disrupts an essential splice site in the poorly characterized gene pla2g12b, which encodes a catalytically inactive phospholipase enzyme with no known function[34]. Although previous studies have suggested that Pla2g12b plays a role in lipid or lipoprotein metabolism[10,33,35–38], it is unclear whether it impacts lipoprotein biogenesis, secretion, intravascular processing, or systemic lipid metabolism. We found that pla2g12b is universally conserved within the vertebrate lineage and absent in invertebrates (Supplementary Fig. 1A), suggesting that the emergence of this gene coincided with the emergence of TRLs at the base of the vertebrate lineage. Further, we found that its expression is enriched in lipoprotein-producing tissues (Supplementary Fig. 1B), but expression is reduced in the fasted state (Supplementary Fig. 1C), consistent with a potential role in lipoprotein metabolism. We therefore sought to define the specific role of Pla2g12b in TRL production.

We used electron microscopy to characterize the subcellular pattern of lipid accumulation in YSL of pla2g12b[sa659] mutants, which revealed numerous lipid micelles bound within an endomembrane compartment (Fig. 1C, Supplementary Fig. 2A–C). A similar pattern of lipid accumulation has been observed in APOB-deficient rats[39], suggesting that Pla2g12b may play a role in transferring lipids to ApoB. Notably, this phenotype differs markedly from the large cytoplasmic lipid droplets (CLDs) observed when Mtp activity is disrupted (Supplementary Fig. 2D). We performed epistasis analyses using the Mtp inhibitor lomitapide, and demonstrated that Pla2g12b acts downstream of Mtp in the TRL biogenesis pathway (Supplementary Fig. 2D).

Intracellular lipid micelles predominantly take the forms of either CLDs, TRLs, or LLDs[18]. We used protein markers (Fig. 1A) to determine which of these three potential particle classes accumulates in the YSL of pla2g12b[sa659] mutants. CLDs are the canonical sites of intracellular lipid storage, which can be specifically labeled with established transgenic reporters such as GFP-tagged Perilipin-2 Fus(EGFP-plin2)[40,41]. ApoB is a specific marker of TRLs and their lipolyzed derivatives, enabling TRLs to be detected via ApoB fusion proteins such as the LipoGlo reporter Fus(apoBb.1-NanoLuc)[10,42]. LLDs are lipid micelles within the ER lumen that lack APOB and no protein has been identified that specifically marks these particles. LLDs are therefore classified as the ApoB-free subset of lipid micelles present within the ER lumen[18] (Fig. 1A).

The EGFP-plin2 reporter was used to characterize CLD accumulation in various zebrafish genetic backgrounds (Fig. 1D). Consistent with previous reports, CLDs are virtually undetectable in wild-type larvae, but are readily apparent in the YSL of mutants with loss-of-

function mutations in Mtp (*mttp*[stl] mutants)[12]. Strikingly, CLDs were virtually undetectable in *pla2g12b*[sa659] mutants despite clear intracellular lipid accumulation (Fig. 1D) suggesting that lipids accumulate in an atypical cellular compartment in these mutants.

We then used density gradient ultracentrifugation (DGUC) to determine the size distribution of both nascent (intracellular) and secreted TRLs using the LipoGlo reporter. Crude lysates (representative of serum lipoproteins) show that *pla2g12b* mutants secrete a lower number of TRLs overall, and that the secreted particles are significantly smaller and denser than those of wild-type controls (Fig. 1E). We reasoned that this phenotype could result from either an inability to produce large TRLs, or defects in the specialized secretory pathway responsible for the secretion of large cargoes such as TRLs[43].

To distinguish between these two possibilities, we analyzed the density profile of TRLs in microsomal (ER-lumenal) extracts as well (Fig. 1F). Wild-type extracts revealed a bimodal distribution of densities, with one peak in the high-density lipoprotein range and a second in the very-low-density range (Fig. 1F, black arrowheads in 3 dpf panel). This distribution is consistent with the two-step model of TRL expansion that involves an initial lipidation step (step 1, Fig. 1F) that forms a dense APOB-containing particle, and a second step (Step 2, Fig. 1F) that rapidly expands the particle core with lipids. By contrast, *pla2g12b* mutants were unable to produce the very-low-density particles seen in wild-type samples (cyan trace, Fig. 1F). These findings suggest that the absence of buoyant TRLs in the circulatory system stems from a fundamental inability to produce large TRLs, rather than retention of large TRLs within the ER.

APOB levels are higher in lumenal extracts of *pla2g12b* mutants from 1–3 days post fertilization (dpf), but drop below wild-type levels by 4 dpf (Fig. 1F). To evaluate whether the elevated levels of ApoB in the ER lumen in the early developmental stages resulted from defects in TRL secretion, zebrafish larvae were exposed to the trafficking inhibitor brefeldin-A (BFA). Disruption of the secretory pathway with BFA led to increased accumulation of ApoB in the ER lumen in both wild-type and *pla2g12b* mutants (Supplementary Fig. 3), suggesting that TRL secretion is intact in both genotypes. Consistent with previous reports, we found that BFA also interfered with TRL expansion[44]. We conclude that BFA disrupts both TRL expansion and cellular trafficking, whereas mutations in *pla2g12b* selectively disrupt TRL expansion while leaving TRL secretion intact.

As abnormally large lipid micelles were observed on electron micrographs, and abnormally small nascent TRLs were detected via DGUC, we concluded that the large lipid micelles accumulating in the YSL of *pla2g12b* mutants must be APOB-free. As these lipid droplets are not labeled by EGFP-plin2 (Fig. 1D) and appear to be bound within a lumenal compartment (Fig. 1C), we hypothesized that these particles may be ER-lumenal lipid droplets (LLDs). We therefore used the fluorescent reporter *ER-tdTomato* to visualize ER morphology in *pla2g12b* mutants and wild-type controls (Fig. 1G). The wild-type ER presented a typical morphology of a network of interconnected tubules with diameters <100 nm. By contrast, the ER was greatly distended in *pla2g12b* mutants, swelling to several microns in diameter and containing large inclusions (white arrowheads, Fig. 1G) resembling those seen in electron micrographs (Fig. 1B). By showing that these lipid droplets lack APOB and PLIN2, and are bound within the ER lumen, we demonstrate that *pla2g12b* mutants accumulate lipids in the form of LLDs.

To investigate whether PLA2G12B plays a similar role in human cells, we used CRISPR/Cas9 to generate loss of function mutations in liver (HepG2) and intestinal (Caco2) derived cell lines. Similar to our live animal studies, human cultured cells lacking functional PLA2G12B show defects in the production of large TRLs, and instead secrete abnormally small lipid-poor lipoproteins (Fig. 1H, I, Supplementary Fig. 4A, C). To further support that mutations in *pla2g12b* interfere with TRL expansion rather than secretion of large cargoes, we demonstrate

that secretion of the large cargo Collagen XII is unchanged (Supplementary Fig. 4E, F). Consistent with our results in larval zebrafish studies and previous work in mice[36–38], we detect decreased overall secretion of APOB in *PLA2G12B*[−/−] mutant cell lines (Supplementary Fig. 4B, D).

In summary, *pla2g12b* mutants produce and secrete abnormally small TRLs, while retaining abnormally large amounts of lipids in the form of LLDs within the ER. These results suggest that Pla2g12b is a key mediator of lipid partitioning between nascent TRLs and LLDs.

## Functional domains of Pla2g12b

PLA2G12B is a catalytically inactive member of the phospholipase gene family[34], and it is unclear what protein domains are responsible for its role in lipid partitioning. We developed a rescue assay that could be used to screen for functional domains using an allelic series. We first demonstrated that microinjection of a plasmid encoding wild-type *pla2g12b* under control of a YSL-specific promoter[45] (Fig. 2A) completely reverses the darkened-yolk phenotype in mutant larvae (Fig. 2B). We then used site-directed mutagenesis to generate a panel of rescue plasmids encoding mutant alleles of *pla2g12b* tagged with either mScarlet or 3xFLAG (Supplementary Table 1), and evaluated the impact of each allele on rescue activity in mutant zebrafish larvae (Fig. 2C).

We used two different strategies to select residues likely to have functional activity. We began by identifying putative functional motifs in silico, including an amino-terminal signal peptide (SP), a carboxyl-terminal ER-retention signal (KD), a conserved calcium-binding motif[34] (CA), and an essential cysteine (C129Y) identified in an ENU screen in mice[36] (EN). For the remainder of the sequence, we used evolutionary conservation to identify the subset of residues most likely to be involved in neofunctionalization[46]. We generated an alignment of Pla2g12b and its catalytically active ohnolog Pla2g12a for human, mouse, and zebrafish protein sequences. We then identified positions that were highly conserved in Pla2g12b but differed from Pla2g12a (Supplementary Fig. 5). A total of 45 amino acid positions met these criteria, which were binned together to generate 12 mutant alleles with an average of 4 amino acid substitutions per allele (Fig. 2C, Supplementary Fig. 5). By integrating evolutionary information from both paralogs and orthologs, we hypothesized that this recoding strategy may represent an efficient method to screen for neofunctionalized positions while minimizing the likelihood of protein misfolding.

Rescue activity was quantified by estimating the percentage of the yolk surface area exhibiting the translucent appearance typical of wild-type yolks (Supplementary Fig. 6). All four alleles that disrupted putative functional domains virtually abolished rescue activity, indicating that these motifs are all essential for protein function (blue labels, Fig. 2C). Four of the alleles generated through paralog recoding abolished rescue activity as well (red labels, Fig. 2C), which we denote as essential domains A and B.

We developed an additional rescue assay in human cells that uses the DGUC to determine the density profile of secreted TRLs as a metric of rescue activity. *PLA2G12B* mutant cell lines produced significantly fewer buoyant TRLs (dashed box, Fig. 2D, E), but this phenotype could be reversed through transfection with wild-type PLA2G12B (PLA) or 3xFLAG-tagged PLA2G12B (FPLA). We selected a subset of alleles to test in the rescue assays and validated that domains essential for rescue in the zebrafish assays were also required for rescue of the TRL expansion phenotype in human cells (Fig. 2F).

We then mapped these functional domains onto the predicted structure of Pla2g12b generated by AlphaFold[47] (Fig. 2G, Supplementary Fig. 7). This 3D structure positions functional domains A and B directly adjacent to each other on a protein surface distal to the calcium-binding domain, but the functional activities of these domains remains unknown.

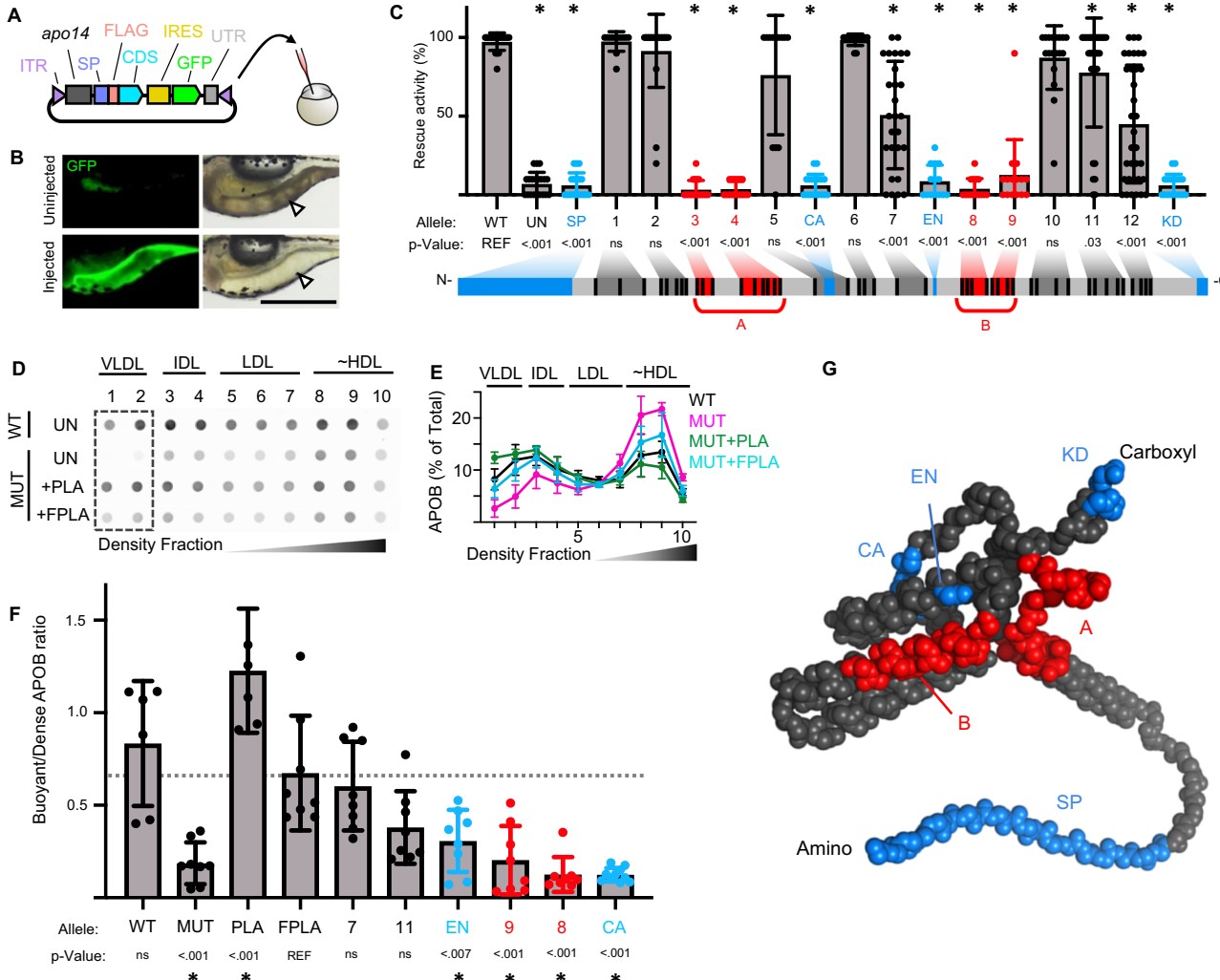

**Fig. 2 | Functional domains of Pla2g12b. A** Schematic representation of rescue plasmid, containing a YSL-specific promoter (apo14), a signal peptide (SP), a 3xFLAG epitope (FLAG), the *pla2g12b* coding sequence (CDS), an internal ribosome entry site (IRES), a GFP reporter, and the endogenous 3′ UTR of *pla2g12b* flanked by inverted terminal repeats (ITR) assembled in the pDestTol2pA backbone to mediate genome integration via Tol2 transposase. **B** Representative fluorescent and brightfield images of *pla2g12b^sa659^* mutants injected with rescue plasmid (Injected) and uninjected controls (Uninjected), showing reversal of the dark-yolk phenotype (white arrowhead) and expression of a GFP marker throughout the YSL of successfully injected larvae. Scale bar = 500 μm. **C** Rescue activity of *pla2g12b* alleles (depicted by shaded blocks) as measured by reversal of the dark-yolk phenotype. 4 mutant alleles of *pla2g12b* were designed that disrupted predicted functional features (blue), and 12 were designed based on evolutionary conservation (gray and red labels). Data are mean ± s.d. of $n \geq 12$ biological replicates for each allele split across at least two independent experiments. One-way ANOVA $p < 0.0001$, the asterisk denotes $p < 0.05$ by Tukey's multiple comparison test. **D** Representative dot blot showing density distribution of APOB in media from cultured HepG2 cells showing that either endogenous or transgenic PLA2G12B is required to efficiently produce buoyant lipoproteins with densities similar to VLDL. **E** Quantification of APOB density distribution from dot blots. Points and error bars represent mean ± s.d. of $n \geq 6$ biological replicates. **F** Rescue activity of various *PLA2G12B* alleles quantified as buoyant/dense ratio, showing consistent findings with zebrafish assays. Bars represent mean ± s.d. of $n \geq 6$ biological replicates. One-way ANOVA, $p < 0.0001$. Asterisks denote $p < 0.01$ by Dunnett's multiple comparison test. **G** Overlay of essential domains identified in the allelic series onto the predicted structure of Pla2g12b generated by AlphaFold, including essential domains A and B (red), an amino-terminal signal peptide (SP), a calcium-binding domain (CA), an essential cysteine residue identified in an ENU screen (EN), and a carboxyl-terminal ER-retention motif (KD). Source data are provided as a Source Data file.

## Membrane and protein associations

Although PLA2G12B is a member of a secreted family of phospholipases, the presence of an amino-terminal signal peptide and carboxy-terminal ER-retention motif (KDEL) suggest that this protein would be retained within the ER. We designed additional alleles of PLA2G12B fused to mScarlet to visualize PLA2G12B localization in vivo. HepG2 cells were transfected with chemically inducible Signal-Peptide-mScarlet-PLA2G12B expression constructs and co-stained with ER-tracker dye, revealing clear colocalization with the ER (Fig. 3A). Sustained expression of this construct under control of a constitutive promoter led to formation of fibrous intracellular

aggregates, which is likely an artifact of overexpression (Supplementary Fig. 8A). Microinjection of similar mScarlet-Pla2g12b expression constructs into zebrafish larvae under control of a YSL-specific promoter revealed that expression was restricted to the YSL, with no evidence of secretion into the bloodstream (Fig. 3B). This observation is consistent with previous analyses of the human serum proteome showing that PLA2G12B is virtually undetectable in the human bloodstream[48,49]. Importantly, the mScarlet-Pla2g12b fusion construct reversed the darkened-yolk phenotype in a dose dependent manner (Fig. 3C), and also displayed a pattern consistent with ER-localization under confocal imaging

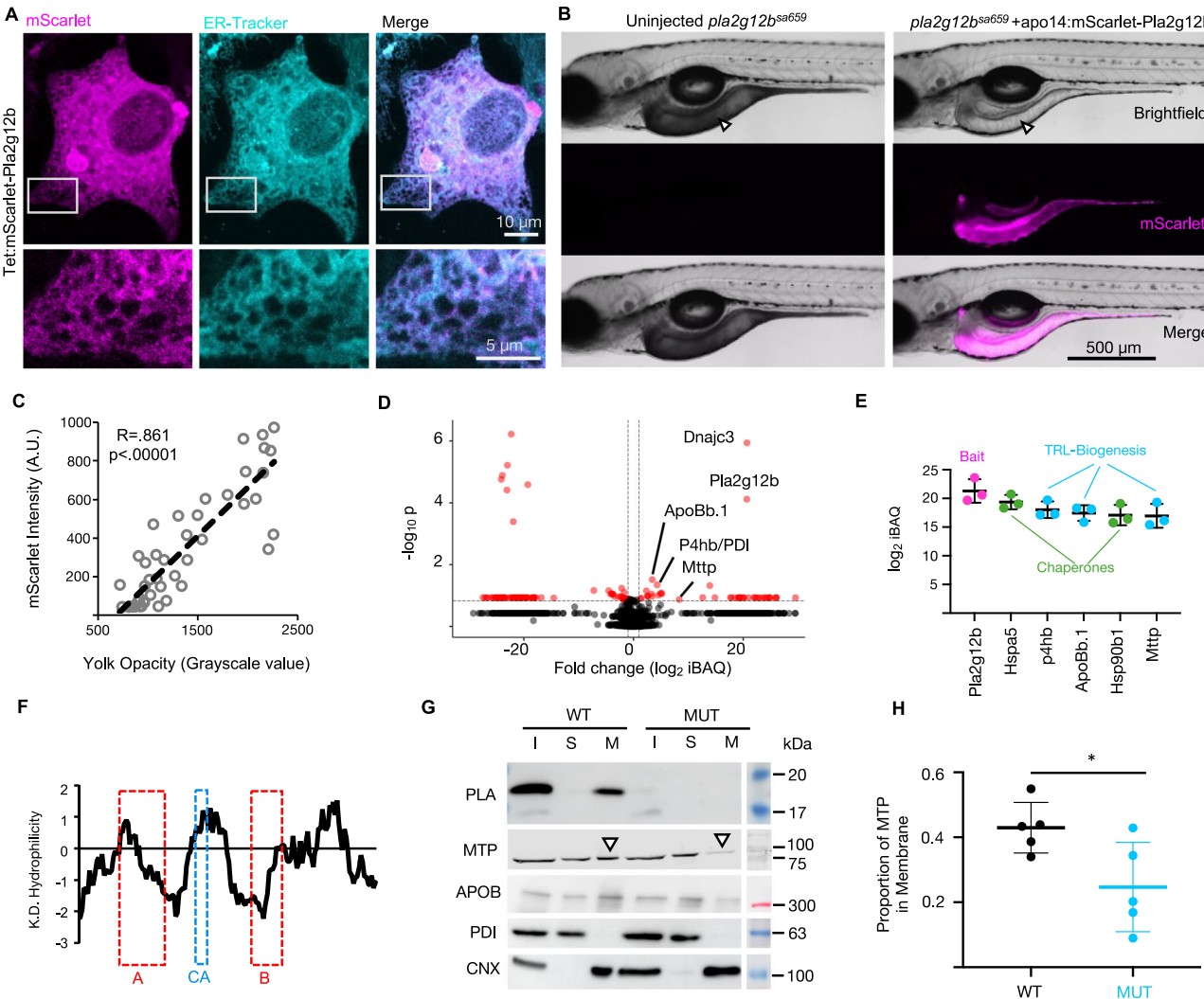

**Fig. 3 | Interactions of Pla2g12b with TRL biogenesis factors and the ER membrane. A** Confocal micrographs of HepG2 cells expressing mScarlet-PLA2G12B fusion proteins, which co-localize with ER-tracker dye. White boxes denote regions of interest that are magnified in lower panels. Scale bars = 10 μm in upper panel, 5 μm in the enlarged lower panel. The image is representative of $n = 10$ biological replicates split across two independent experiments. **B** Micrographs of *pla2g12b^sa659* mutants injected with mScarlet-Pla2g12b expression constructs driven by a YSL-specific promoter (*apo14*). Transgenic expression of the Pla2g12b fusion protein reverses the dark-yolk phenotype (white arrowhead), suggesting the protein is functional and retained within the ER. Scale bar = 500 μm. **C** Correlation between mScarlet signal intensity and grayscale value of the YSL. Pearson Correlation Coefficient = 0.861, $n = 47$, and $p < 0.00001$ by two-tailed Pearson's $r$ test. **D** Volcano plot depicting the significantly enriched peptides detected in unbiased co-IP/MS assays pull-downs against 3xFLAG-tagged Pla2g12b. Data points represent unpaired and uncorrected two-tailed $t$ tests from $n = 3$ biological replicates. **E** The top 6 peptides passing statistical and fold-change thresholds ranked by abundance (iBAQ). 3xFLAG-tagged Pla2g12b (magenta) served as the bait protein, and the most abundant interacting partners included chaperone proteins (green) and proteins involved in TRL biogenesis (cyan). Data are mean ± s.d. of $n = 3$ independent experiments. **F** Kyte-Dolittle (K.D.) hydrophilicity plot of Pla2g12b amino acid sequence illustrating that the A and B domains are present in hydrophobic regions of the protein, whereas the calcium-binding domain (CA) is in a hydrophilic stretch. **G** Representative Western blots of input (I), soluble (S), and membrane (M) cellular fractions of HepG2 cells show that Pla2g12b partitions exclusively to the membrane fraction, and that less MTP is recruited to the membrane fraction in cells lacking functional PLA2G12B (white arrowhead). Protein disulfide isomerase (PDI) and Calnexin (CNX) are shown as markers of soluble and membrane fractions, respectively. **H** Quantification of MTP partitioning between soluble and membrane fractions as shown in **G**. Data are from $n = 5$ biological replicates. Unpaired two-tailed $t$ test, $p = 0.03$. Source data are provided as a Source Data file.

(Supplementary Fig. 8B, C). Further, Western blots demonstrated that PLA2G12B is undetectable in the media of human cultured cells (Supplementary Figs. 4E, F and 9). These findings demonstrate that PLA2G12B is not secreted, but instead is retained within the ER.

We hypothesized that identifying the interacting partners of Pla2g12b could provide further insight into its sub-organellar localization and mechanism of action. We injected zebrafish larvae with the functional FLAG-tagged wild-type rescue plasmid and performed co-immunoprecipitation and mass spectrometry (co-IP/MS) against the highly specific 3xFLAG epitope linked to Pla2g12b. Peptides were filtered for fold-enrichment and statistical significance and then ranked by abundance (Fig. 3D, Supplementary Fig. 10). The top five most abundant proteins meeting these thresholds (Fig. 3E) included the core components of the TRL assembly pathway (ApoB and the two Subunits of Mtp (Mttp and Pdi/P4hb)), as well as various chaperone proteins.

As both Mtp and ApoB interact with lipid membranes[50], we assessed whether Pla2g12b may have membrane-binding properties as well. We analyzed the hydrophilicity of the primary amino acid sequence and found that the essential A and B domains are predominantly hydrophobic (Fig. 3F). We then performed Western blots on cells fractionated into soluble and membrane components and

found that Pla2g12b is present exclusively in membrane fractions (Fig. 3G).

Given that Pla2g12b interacts extensively with both the ER membrane and the proteins involved in TRL biogenesis, we hypothesized that it may concentrate components of the TRL assembly pathway along the ER membrane. Although the cellular MTP content remains unchanged (Supplementary Fig. 9), a significantly lower proportion of MTP was associated with the membrane in $PLA2G12B^{-/-}$ mutant HepG2 cells (white arrowheads, Fig. 3G, H). A similar trend was observed when mutant cells were transfected with a $PLA2G12B$ allele lacking a functional calcium-binding domain, but did not meet the threshold of significance (Supplementary Fig. 11).

We conclude that PLA2G12B is retained within the cell where it associates tightly with the ER membrane as well as numerous key players in TRL biogenesis and concentrates the lipid transfer protein MTP along the lumenal face of the ER membrane. As the ER membrane is a site of active TRL biogenesis, particularly within the YSL, additional studies will be required to distinguish direct protein-protein interactions from incidental interactions resulting from colocalization along the ER membrane.

### Calcium enhances lipidation of TRLs

Many phospholipase enzymes have calcium-dependent catalytic activity[51,52]. Although the phospholipase activity of PLA2G12B has been lost[53,54], we found that its calcium-binding domain remains conserved and essential for function[53,54] (Fig. 2C, F). We therefore sought to investigate the role of calcium in lipoprotein expansion using a combination of in vivo and in vitro assays.

Thapsigargin (THAP) depletes calcium within the ER by interfering with the sarcoplasmic/ endoplasmic reticulum $Ca^{2+}$ ATPase (SERCA) calcium pump[55]. Exposure to THAP caused wild-type larvae to accumulate a pool of unusually small TRLs, effectively phenocopying the TRL expansion defect observed in $pla2g12b^{sa659}$ mutants (Fig. 4A). Nocodazole (NOC) is a microtubule depolymerizing agent that disrupts intracellular trafficking[56], which we found to significantly increase the yield of large TRLs in microsomal extracts by inhibiting TRL secretion (Fig. 4B). Simultaneous exposure to both inhibitors resulted in a more pronounced accumulation of abnormally small TRLs in wild-type larvae (Fig. 4C). These findings demonstrate that calcium is essential for TRL expansion in vivo.

Triglyceride transfer activity can be monitored directly in vitro[9]. This assay monitors the increase in fluorescence intensity as MTP transfers fluorescently labeled triglycerides from quenched donor vesicles to unquenched acceptors[9,57], either TRLs or APOB-free acceptor vesicles (AV, Fig. 4D). Soluble ER-lumenal extracts from wild-type and $Pla2g12b^{hlb218}$ mutant mice were used as a source of MTP for these assays. Transfer activity to each of these donor types was evaluated in the context of varying concentrations of calcium. Calcium supplementation had no effect on the rate of triglyceride transfer to APOB-free acceptor vesicles (Fig. 4E), but increased the rate of triglyceride transfer to TRLs by ~60% (Fig. 4F, G). This in vitro assay uses only soluble ER-resident proteins, which excludes the PLA2G12B (Fig. 3G). Accordingly, identical results were observed when $PLA2G12B^{hlb218}$ extracts were used as a source of MTP (Fig. 4H–K), which suggests that the calcium dependence of TRL expansion is not solely attributable the calcium dependence of PLA2G12B.

We conclude that under in vitro conditions, high concentrations of calcium selectively increase triglyceride transfer to TRLs but have no effect on transfer to APOB-free acceptor vesicles (Fig. 4K). We also demonstrate that physiological calcium levels are essential for TRL expansion in vivo (Fig. 4A–C).

### Hypolipidemia in $Pla2g12b^{-/-}$ mutants

Mutations in $Pla2g12b$ result in significantly lower levels of serum triglycerides and cholesterol in mice[36]. To determine the precise changes in lipoprotein composition contributing to hypolipidemia, we performed FPLC and enzymatic lipid profiling on plasma isolated from $Pla2g12b^{hlb218}$ mutant mice and their wild-type siblings (Fig. 5A–H). $Pla2g12b^{hlb218}$ is a point mutation generated by ENU that substitutes a cysteine residue at position 129 with tyrosine (C129Y)[36]. As expected, we found that VLDL, the triglyceride-rich lipoproteins secreted from the liver, exhibited a >80% reduction in triglyceride content (Fig. 5B). However, despite this drastic decrease in triglyceride content, the cholesterol content of VLDL particles remained unchanged (black arrowhead Fig. 5E, F). We conclude that the abnormal TRLs secreted by $Pla2g12b^{hlb218}$ mutants are selectively depleted in triglycerides.

Serum triglyceride and cholesterol levels are key determinants of atherosclerotic cardiovascular disease risk[23]. We hypothesized that the hypolipidemia observed in $Pla2g12b^{hlb218}$ mutants may result in resistance to atherosclerosis. We used an established mouse model of atherosclerosis[58]: feeding of the western diet in the context of adeno-associated virus (AAV) induced overexpression of Proprotein convertase subtilisin/kexin type 9 (PCSK9). PCSK9 promotes degradation of the low-density lipoprotein receptor (LDLR), resulting in reduced clearance of atherogenic TRLs and their lipolyzed byproducts from the circulatory system[59]. $Pla2g12b^{hlb218}$ mutant mice and their wild-type siblings were injected with a single dose of PCSK9-AAV at 4 months of age and fed a western diet for 6 months following injection (Fig. 5I).

This hyperlipidemia paradigm significantly increased serum levels of triglycerides and cholesterol in both wild-type and mutant mice, but serum lipid levels remained significantly lower in $Pla2g12b^{hlb218}$ mutants (Fig. 5J, K). Notably, while this hyperlipidemia paradigm caused cholesterol levels to increase dramatically in $Pla2g12b$ mutant mice (>100-fold increase), the increase in serum triglyceride levels was much less pronounced (~7-fold increase). Mice were euthanized 6 months post-injection and major arteries were inspected for signs of atherosclerosis. Atherosclerotic lesions were readily apparent in the aortic arches of wild-type mice, but were virtually undetectable in $Pla2g12b^{hlb218}$ mutants (white arrowhead, Fig. 5L). En face Oil-Red-O staining was used to detect atherosclerotic lesions more sensitively (Fig. 5M), which confirmed that lesions were significantly more abundant in wild-type mice.

We performed additional metabolic profiling of $Pla2g12b^{hlb218}$ mutant mice and $pla2g12b^{sa659}$ mutant zebrafish, and found that they are viable and fertile, exhibit similar growth rates, body compositions, and activity levels to wild-type controls (Supplementary Fig. 12). Consistent with our observations in the YSL and previous reports[36], we did observe intracellular lipid accumulation in lipoprotein-producing tissues of adult zebrafish (Supplementary Fig. 12D, E). Loss of PLA2G12B function therefore supports relatively normal growth and metabolism despite intestinal and hepatic lipid accumulation and profound serum hypolipidemia and resistance to atherosclerosis.

Further, we identified two independent genome-wide association studies[60,61] that associated human polymorphisms linked to the $PLA2G12B$ promoter with plasma triglyceride and cholesterol levels[62] (Supplementary Fig. 13). We found that these variants are located in the first intron of $PLA2G12B$, and overlap with a variety of open-chromatin marks and predicted transcription factor binding sites. Genetic variants linked to $PLA2G12B$ are therefore significantly associated with serum triglyceride levels in humans.

## Discussion

Here we use zebrafish, murine, human, and in vitro systems to establish $PLA2G12B$ as the key driver of TRL expansion within the vertebrate lineage. We demonstrate that PLA2G12B mutants secrete lipid-poor TRLs and produce an excess of LLDs (Fig. 1), annotate functional protein domains within PLA2G12B (Fig. 2), and show that it interacts with the ER membrane and canonical components of the lipoprotein assembly pathway (Fig. 3). The highly conserved calcium-binding domain of PLA2G12B led us to perform a thorough investigation of the

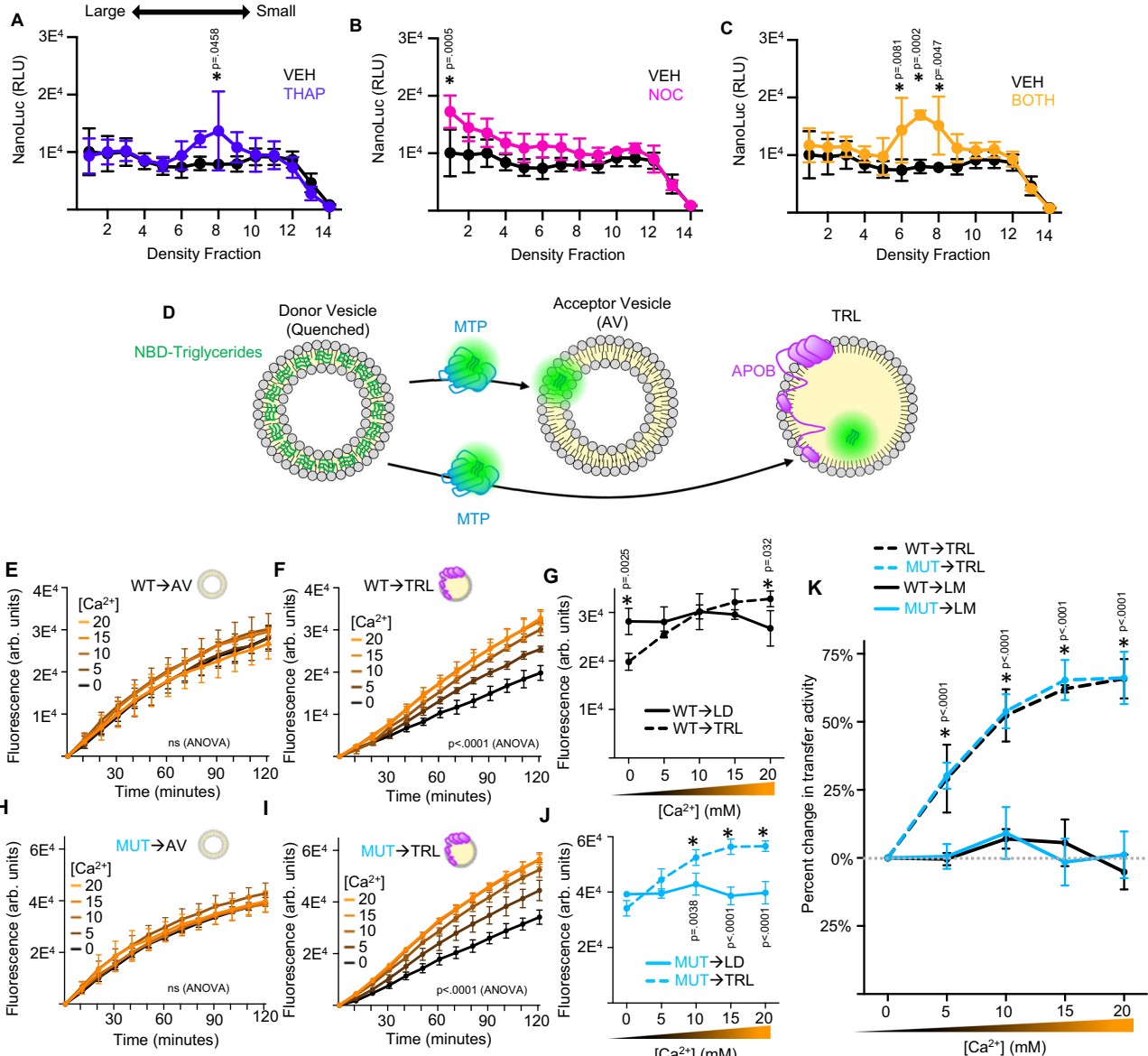

**Fig. 4 | Calcium is essential for TRL expansion in vivo and in vitro. A** Density gradient profiles of TRLs from wild-type 2 dpf zebrafish larvae exposed to thapsi-gargin (THAP) to deplete ER-calcium levels. THAP causes accumulation of a pool of unusually small lipoproteins, whereas **B** nocodazole (NOC) can be used to capture large TRLs more efficiently. **C** Co-exposure to NOC and THAP (BOTH) causes more pronounced accumulation of dense TRLs. Data are mean ± s.d. of $n = 3$ biological replicates. Two-way ANOVA $p < 0.005$ for interaction for all treatments, asterisk denotes $p < 0.05$ by Šídák's multiple comparison test. **D** Schematic illustration of in vitro triglyceride transfer assay which quantifies transfer of fluorescently labeled (NBD) triglycerides from quenched donor vesicles to unquenched acceptor vesicles (AV) or TRLs. **E** The rate of triglyceride transfer is unaffected by calcium concentration when microsomal extracts from wild-type mice are used to transfer triglycerides to APOB-free acceptor vesicles. Data are mean ± s.d. of $n = 3$ technical replicates. Two-way ANOVA $p = 0.57$. **F** Transfer to APOB-containing TRLs is significantly enhanced by calcium supplementation. Data are mean ± s.d. of $n = 3$ technical replicates. Two-way ANOVA $p < 0.0001$ for interaction. **G** Comparison of endpoint fluorescence for different donors under varying calcium conditions, showing that transfer to TRLs becomes more favorable in high-calcium conditions. Data are mean ± s.d. of $n = 3$ technical replicates. Two-way ANOVA $p < 0.001$ for interaction, asterisk denotes $p < 0.05$ by Šídák's multiple comparison test. **H–J** Microsomal extracts from *PLA2G12B*$^{hlb218}$ mice exhibited identical patterns of calcium dependence compared to wild-type extracts. Data are mean ± s.d. of $n = 3$ technical replicates. Two-way ANOVA $p < 0.0001$ for interaction, asterisk denotes $p < 0.005$ by Šídák's multiple comparison test. **K** Comparison of relative changes in transfer activity resulting from calcium supplementation in different genotypes and donor types, showing that calcium selectively enhances transfer to APOB-containing TRLs irrespective of *PLA2G12B* genotype. Data are mean ± s.d. of $n = 3$ technical replicates. Two-way ANOVA $p < 0.001$ for interaction, asterisk denotes $p < 0.05$ by Šídák's multiple comparison test. Elements in **D–F**, **H** and **I** have been adapted from previous work[10] under Creative Commons Attribution 4.0 International License. Source data are provided as a Source Data file.

role of calcium in TRL expansion, which revealed an essential role of calcium in lipid partitioning in vitro and in vivo (Fig. 4). Finally, we evaluated the physiological consequences of PLA2G12B disruption, showing that *Pla2g12b*$^{hlb218}$ mutants exhibit profound hypolipidemia and resistance to atherosclerosis coupled with otherwise normal growth and metabolism (Fig. 5).

These observations demonstrate that the acquisition of triglyceride-transfer activity by MTP was necessary, but not sufficient, to drive the emergence of TRLs in the vertebrate lineage. We identify PLA2G12B as a vertebrate-specific gene that ensures efficient lipidation of TRLs and restricts lipid accumulation in LLDs (Fig. 6). Mechanistically, we find that PLA2G12B increases the proportion of MTP

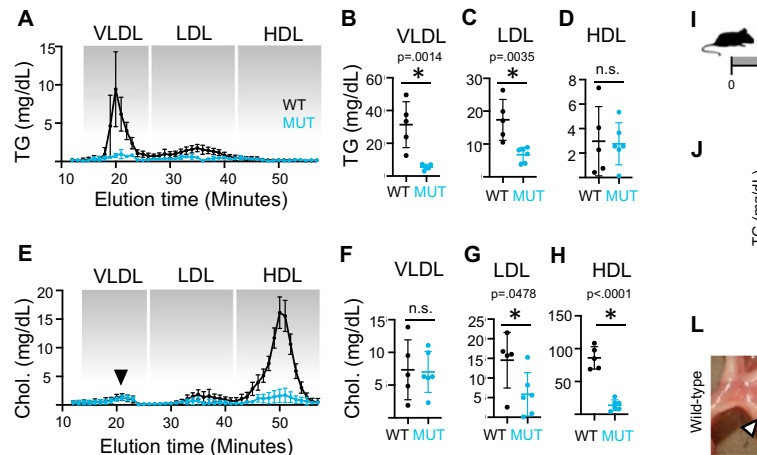

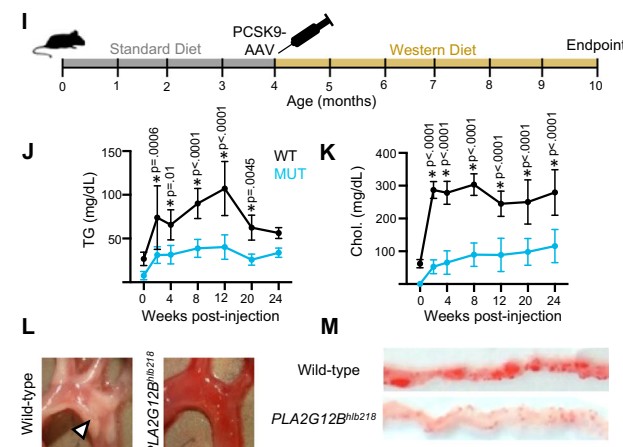

**Fig. 5 | PLA2G12B mutants are resistant to hyperlipidemia and atherosclerosis.**
**A** Fast-protein liquid chromatography (FPLC) was used to fractionate different classes of lipoproteins from 4-month-old wild-type (black) and mutant (cyan) mouse plasma, and enzymatic assays were used to determine the triglyceride content of each class. Points and error bars represent mean ± s.d. of $n ≥ 5$ biological replicates. **B** Triglyceride levels were significantly lower in the VLDL and **C** LDL fractions of *Pla2g12b^hlb218* mutants, and **D** triglyceride levels were vanishingly low in HDL particles, which are inherently triglyceride-poor. Individual data points are shown, with $n ≥ 5$ for all comparisons. Bars reflect mean ± s.d., and the asterisks denote \**p* < 0.05, \*\**p* < 0.01, and \*\*\*\**p* < 0.0001 by unpaired two-tailed *t* test. **E** Enzymatic measurement of cholesterol levels in FPLC fractions revealed that **F** the cholesterol content of the VLDL fraction was surprisingly unaffected by mutations in *Pla2g12b*, despite significant reductions in the cholesterol content of **G** LDL and **H** HDL classes. Individual data points are shown, with $n ≥ 5$ for all comparisons. Bars reflect mean ± s.d., and the asterisks denote \**p* < 0.05, \*\**p* < 0.01, and \*\*\*\**p* < 0.0001 by unpaired two-tailed *t* test. **I** Schematic illustration of hyperlipidemia paradigm

involving injection of adult mice with PCSK9-AAV at 4 months of age followed by 6 months of high-fat western diet. **J** Quantification of plasma levels of triglycerides and **K** cholesterol following AAV injection and transition to Western diet, where time 0 corresponds to samples collected immediately prior to induction of hyperlipidemia. Plasma lipid levels increased significantly in both genotypes, but hyperlipidemia was much more pronounced in wild-type samples. Data are mean ± s.d. of $n ≥ 5$ gender-balanced samples per group. Two-way ANOVA $p < 0.005$ for interaction at all stages, the asterisk denotes $p < 0.05$ by Šídák's multiple comparison test. **L** Brightfield images of aortic arches dissected at experimental endpoints, showing widespread accumulation of atherosclerotic lesions (white arrowhead) in wild-type mice, whereas lesions were virtually undetectable in *Pla2g12b^hlb218* mutants. **M** *En face* oil-red-O staining of dissected aortas, demonstrating higher lipid accumulation (red) in wild-type arteries. All data represent pooled samples from a gender-balanced cohort of males and females. Source data are provided as a Source Data file.

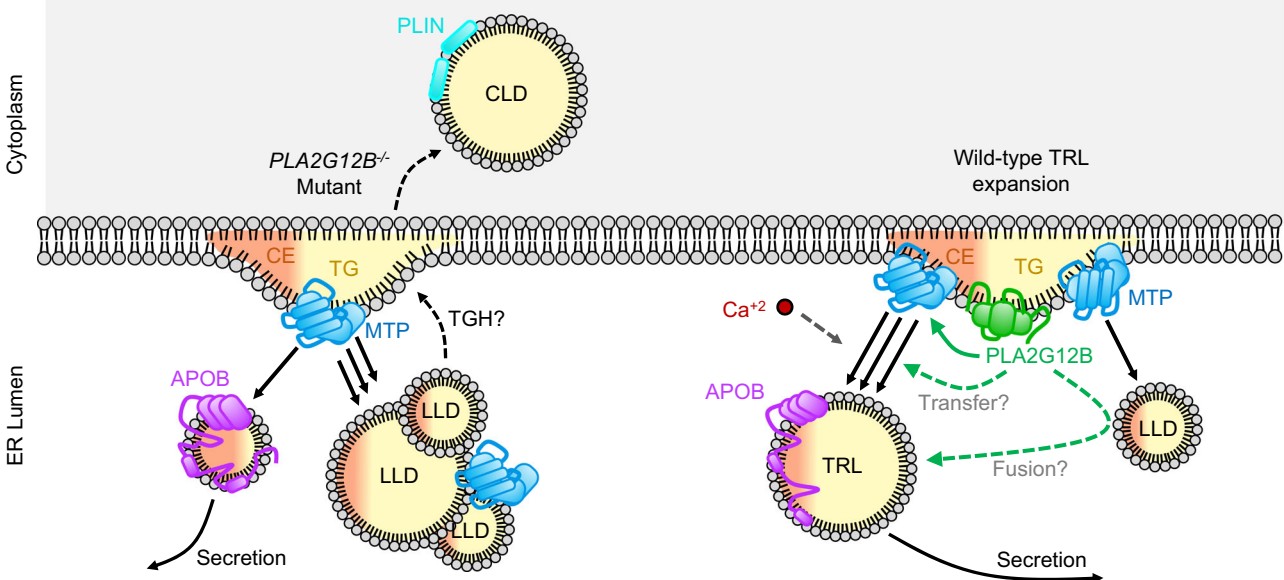

**Fig. 6 | Role of PLA2G12B in TRL expansion.** In the absence of functional PLA2G12B (left portion of schematic), MTP transfers only a small quantity of lipids, predominantly cholesteryl-esters (CE), to nascent lipoproteins. Triglycerides are transferred preferentially to LLDs, and eventually accumulate in CLDs in adult tissues. In the presence of PLA2G12B (right portion of schematic), a higher proportion of MTP is associated with the ER membrane (solid green arrow). PLA2G12B modulates the partitioning of lipids between TRLs and LLDs and ensures efficient

channeling of lipids to nascent TRLs either by enhancing direct transfer of lipids from the ER membrane (upper dashed green arrow) or via transfer-independent mechanisms such as fusion of LLDs with TRLs (lower dashed green arrow). Calcium is also essential for the proper expansion of nascent TRLs, although its specific role in this process remains unclear (dashed gray arrow). Elements of this panel have been adapted from previous work[10] under Creative Commons Attribution 4.0 International License.

bound to the ER membrane (Fig. 3G, H). As the initial stages of APOB lipidation occur along the ER membrane[6], it is possible that concentration of MTP along this two-dimensional surface contributes to enhanced lipidation of APOB. Alternatively, other catalytically inactive phospholipases have been shown to have membranotropic properties[54], which led us to speculate that membrane remodeling by PLA2G12B could also contribute to enhanced lipidation of TRLs. Finally, several lines of evidence have suggested that TRL expansion occurs through fusion between TRLs and LLDs[16,17,21,44]. If this fusion model is correct, then it would suggest that PLA2G12B enhances TRL expansion by promoting fusion between TRLs and LLDs. Our results are therefore perfectly consistent with both prevailing models of TRL expansion; PLA2G12B may enhance TRL expansion by promoting transfer of lipids from the ER membrane directly to TRLs, or lipids may be transferred to an LLD intermediate prior to being incorporated into TRLs (Fig. 6).

Although PLA2G12B was previously thought to be a secreted protein, this annotation lacks experimental support and appears to be based almost exclusively on the close evolutionary relationship between *pla2g12b* and other secreted phospholipase A₂ genes[35]. We provide experimental evidence supporting that PLA2G12B is actively retained within the ER in all model systems tested, and is not detectable in the serum or media. Although trace amounts of PLA2G12B have been detected in the human bloodstream using mass spectrometry[49], this sensitive technique routinely identifies canonical intracellular proteins that leak into the bloodstream. We therefore provide experimental evidence regarding the localization of Pla2g12b, and find that it is not actively secreted, and instead localizes within the ER.

We detect reduced secretion of APOB in all model systems tested, and a transient increase in lumenal ApoB levels in *pla2g12b* mutant larval zebrafish. However, secretion of other factors, including large cargoes such as collagen that require specialized secretory machinery, remains intact[43]. Further, we found that pharmacological inhibition of the secretory pathway with BFA increased levels of lumenal ApoB in *pla2g12b* mutants. These findings suggest that although ApoB secretion is lower in *pla2g12b* mutants, Pla2g12b is not required for TRL secretion. Instead, our results suggest that the defects in TRL secretion may be a secondary consequence of improper TRL expansion, which could compromise interactions with sorting receptors required for TRL secretion[63] or activate quality control machinery that degrades defective TRLs[3,7,64].

Transmembrane 6 superfamily 2 human (TM6SF2) has recently been identified as a factor involved in TRL expansion[30], similar to PLA2G12B. Curiously, *tm6sf2* does not appear to be expressed in the zebrafish YSL[65], nor was TM6SF2 protein detectable in our pull-downs against FLAG-tagged PLA2G12B (Fig. 3). These observations call for a thorough analysis of TRL production in the YSL of *tm6sf2* mutants to conclusively determine whether large TRLs can be produced in the absence of TM6SF2.

Although we initially investigated the role of calcium in TRL biogenesis because of the calcium-binding motif present in PLA2G12B, our in vitro assays demonstrated that calcium greatly enhances lipidation of APOB in the absence of PLA2G12B (Fig. 4). As the lipid transfer activity of MTP is unaffected by calcium supplementation when APOB-free acceptor vesicles are used in the transfer assay, the most parsimonious interpretation is that calcium interacts with the APOB and enhances its ability to serve as a lipid acceptor. This hypothesis is in line with previous reports that APOB binds calcium[66], and that MTP transfer to TRLs can be activated by calcium[67]. However, the precise role of the calcium-binding domain of PLA2G12B, and the broader role of calcium in TRL expansion warrant further investigation.

Knockout studies in mice have previously reported that cytoplasmic lipid droplets accumulate in lipoprotein-producing tissues of *PLA2G12B* mutants[37,38]. This contrasts sharply with our findings in the YSL of *pla2g12b* mutant zebrafish, which are devoid of CLDs and instead accumulate LLDs. Consistent with previous studies in mice, we found that adult tissues in zebrafish also accumulate CLDs rather than LLDs (Supplementary Fig. 12D, E). We therefore speculate that differences in the stage of development, rate of TRL production, and levels of lumenal lipase expression (such as Triacylglycerol hydrolase, TGH/Ces3)[18,68] may drive differences in CLD and LLD lipid partitioning between the YSL and adult tissues. However, the observation of LLDs in the YSL of *pla2g12b* mutant zebrafish demonstrates that Pla2g12b is not essential for the transfer of lipids into the ER lumen.

The emergence of TRLs marked a transitional moment in animal evolution. The small, triglyceride-poor lipoproteins produced in invertebrates transport phospholipids and cholesteryl-esters, which function primarily in membrane synthesis and cellular signaling. By contrast, the incorporation of triglycerides transformed lipoproteins into key carriers of metabolic fuel. Triglyceride utilization has become a central aspect of vertebrate metabolism but was accompanied by an unforeseen tradeoff. TRLs are the causative agents of atherosclerotic cardiovascular disease[23], which is among the leading causes of death worldwide[25].

It has been proposed that selectively interfering with the triglyceride-transfer activity of MTP may provide therapeutic value by allowing essential lipids to flow between tissues in triglyceride-poor particles, which may be less likely to induce serum hyperlipidemia and cardiovascular disease[12,69]. Our assessment of cardiovascular disease in *PLA2G12B* mutants provides experimental evidence that interfering with TRL expansion is atheroprotective. However, the diminished triglyceride content of secreted TRLs leads to lipid retention in lipoprotein-producing tissues like the intestine and liver, suggesting that other dietary or pharmacological interventions may be warranted to limit the effects of steatosis. However, our broad panel of metabolic phenotyping suggested that PLA2G12B mutant animals were remarkably healthy despite evidence of steatosis (Supplementary Fig. 12), suggesting it may provide an advantageous tradeoff in certain contexts. The human GWAS data showing that noncoding polymorphisms in putative cis-regulatory regions at the *PLA2G12B* locus are associated with circulating triglyceride and cholesterol levels in humans suggests that modulating the level of PLA2G12B may be a strategy to produce a less atherogenic plasma lipid profile without the steatosis associated with a null allele.

Modulation of PLA2G12B activity may have particular therapeutic value for the treatment of familial lipase deficiencies, such as lipoprotein lipase (LPL) mutants[70,71]. These deficiencies do not respond well to standard therapies, which predominantly act by enhancing lipoprotein uptake, but should be treatable through modulation of lipid secretion via PLA2G12B. Specifically, we show that *Pla2g12b*[hlb218] mutant mice with compromised lipoprotein uptake resulting from PCSK9 overexpression still maintain relatively normal levels of plasma TG (Fig. 5J). While pharmacological inhibition of protein-protein interactions is notoriously difficult, our identification of an essential calcium-binding pocket within PLA2G12B highlights a druggable target domain for rational drug development. PLA2G12B therefore represents a potential therapeutic target to combat the growing global prevalence of hyperlipidemia and cardiovascular disease.

In conclusion, although PLA2G12B is a deeply conserved member of the secreted phospholipase gene family, it does not act as a phospholipase, nor is it secreted. Although numerous genes have been identified that affect vertebrate lipoprotein homeostasis by modulating lipid availability[72], participating in subcellular trafficking[28,63], or mediating intravascular processing and turnover of lipoproteins[73,74], PLA2G12B represents class of proteins that mediates lipid partitioning between LLDs and nascent TRLs.

## Methods

### Ethics statement

All research complies with all ethical guidelines, and approved by the Carnegie Institution Animal Care and Use Committee (Protocols #139 and #156) and NYU Langone Institution Animal Care and Use Committee (Protocol #201900083).

### Animal husbandry and maintenance

**Zebrafish.** Adult zebrafish were maintained on a 14 h light−10 h dark cycle and fed once daily with ~3.5% body weight of Gemma Micro 500 (Skretting USA). All genotypes were bred into the wild-type AB background. The *pla2g12b^sa659^* allele was generated via N-ethyl-N-nitrosourea (ENU) mutagenesis[33], and results in substitution of an adenine nucleotide with thymine within the splice acceptor site of intron 3–4 (g.10194 A > T). Disruption of this splice acceptor site is predicted to result in failure to incorporate the final exon (exon 4) into the mature transcript and cause nonsense-mediated decay[75]. The *mttp^stl^* allele is a point mutation that substitutes a Leucine at position 475 with Proline (L475P)[76]. The *Fus(EGFP-plin2)*[40], *Fus(apoBb.1-NanoLuc)*[10] reporters have been published previously. As zebrafish sex cannot be determined during the larval stages, sex can be excluded as a variable for experiments involving zebrafish larvae. All procedures comply with all relevant ethical regulations and were approved by the Carnegie Institution Animal Care and Use Committee (Protocol #139).

**Mice.** All mice used in this study were derived from incrosses of *Pla2g12b^hlb218/+^* mice purchased from the Jackson Laboratory (Strain #:008508), or incrosses of their heterozygous progeny, which were maintained in the C57BL/6 J background. This allele was generated via ENU mutagenesis, and results in a missense substitution (C129Y). Adult mice were maintained on a 12 h light−12 h dark cycle between 66−70 °F and 30−70% humidity in cages containing ¼" Corn Cob bedding (Envigo, #7097). Mice were provided ad libitum access to water and commercially available LM-485 laboratory mouse diet (Envigo, #7012). Mice were housed with littermates whenever possible with densities below 5 adult mice per cage. All procedures comply with all relevant ethical regulations and were approved by the Carnegie Institution Animal Care and Use Committee (Protocol #156). Atherosclerosis assays were performed on mice housed in a second colony at NYU Langone, where mice were maintained on a 12 h light−12 h dark cycle in cages containing the Andersons Lab Animal Bedding - 1/4" Bed o'Cobs (#AND4B). Mice were provided *ad libitum* access to water and commercially available PicoLab Rodent Chow Diet (#5053) or Envigo Western Diet (#TD.88137) as per experimental requirements. Mice were housed with littermates whenever possible, with densities below 5 adult mice per cage. All procedures comply with all relevant ethical regulations and were approved by the NYU Langone Institution Animal Care and Use Committee (Protocol #201900083).

**Tissue culture.** HepG2 cells (HB-8065 from ATCC) were cultured in DMEM 4.5 g/L Glucose with UltraGlutamine (Lonza, #H3BE12-604F/U1) medium containing 10% fetal calf serum (Gibco, #10270106), supplemented with 100 units/mL penicillin, and 100 µg/mL streptomycin (Invitrogen, #15070063). Cells were incubated at 37 °C in a humidified incubator supplied with 5% $CO_2$. Cells were passaged with 0.25% trypsin after reaching 80% confluence. HepG2 cells were seeded in 6-well plates at a density of $5 \times 10^4$ cells/well and grown for two weeks with media replacement every two days. Caco2 cells (HTB-37 ™ from ATCC) were grown in the same conditions but in DMEM with 20% FCS, and seeded in Transwell 6-well format dishes (PET, 3.0 µm, Cultek, #45353091), and allowed to differentiate for 21 days under these conditions with changes to fresh medium every 3 days.

**Transfection.** For live microscopy, HepG2 cells ($2.5 \times 10^5$ cells per dish) were grown on 35-mm polymer-bottom dishes (Ibidi) with regular DMEM+PenStrep+10% FBS. 24 h later, cells were transfected with plasmid encoding for Tet-ss-mScarlet-PLA2G12B fusion proteins, with 1:1 ratio (DNA plasmid:Lipofectamine) using Lipofectamine 3000 reagent (Thermo Fisher Scientific) following the manufacturer's recommendations. After 48 h, cells were incubated with 0.2 µg/mL doxycycline for 4 h to express the protein. After this time, cells were washed twice with PBS1X, and incubated in Opti-MEM supplemented with ER-Tracker™ Green (BODIPY™ FL Glibenclamide; Thermofisher) following manufacturer's recommendation. Samples were imaged at 37 °C in 5% $CO_2$ atmosphere. High-resolution images were acquired in an inverted Leica SP5 confocal microscope. Transfection efficiencies for these assays generally ranged between 20−50%.

For generation of stable HepG2 cell lines for rescue/complementation assays, The cDNA encoding for wild-type and mutant *pla2g12b* alleles were subcloned into pL309-GFP Lentiviral transfer plasmid[77]. The transgene sequence is followed by an IRES and the coding sequence of GFP to allow selection of infected cells. Lentiviral particles were generated by co-transfecting 1 µg of the transfer plasmid (L309-GFP) containing the *pla2g12b* allele of interest with the packaging plasmids pRSV−REV, pMDLg/pRRE and envelope plasmid VSV-G (1 µg in total), with 6 µl of TransIT reagent (Mirus Bio. Cat. No.: 293) into $0.8 \times 10^6$ HEK−293 T (ATCC, negative for mycoplasma) cells plated in a 60 mm culture dish containing 3 ml of complete media. The following day the cell culture media was changed with fresh media. Two and three days after, the cell media was collected and filtered through a 0.45 µm membrane. $0.8 \times 10^6$ PLA2G12B$^{-/-}$ HepG2 cells were plated in 60 mm dishes. The following day the media was changed for 2 ml of fresh complete media, 1 ml of the filtered viral particles and 8 µg/ml of hexadimethrine bromide (polybrene) (Sigma-Aldrich. Cat. No.: 107689). The following day media was change with fresh media. Two days after, cells expressing GFP were selected by Fluorescent Activated Cell Sorting to ensure that every cell expressed the desired *pla2g12b* allele.

**Light microscopy.** Zebrafish larvae were dechorionated manually using forceps prior to imaging when necessary. Larvae were anesthetized in tricaine and mounted laterally in a solution of 3% methylcellulose (Sigma, M0387) dissolved in embryo medium. Brightfield images were collected using trans-illumination on a Nikon SMZ1500 microscope with HR Plan Apo 1x WD 54 objective set to 3× objective magnification using an Infinity 3 Lumenera camera and Infinity Analyze 6.5 software.

**Fluorescence microscopy.** Fluorescent images of whole-larvae were collected on a Zeiss Axiozoom v16 microscope set to ×30 magnification equipped with a Zeiss AxioCam MRm. Confocal images of zebrafish larvae were taken with a Leica DMI6000 inverted microscope using a Leica 63x objective with a Leica TCS-SP5 II confocal scanner. The mCherry fluorophore was excited with a 561 nm laser and detected via photomultiplier tubes with a collection window set to 610−650 nm. The EGFP fluorophore was excited with a 488 nm laser and detected via photomultiplier tubes with a collection window set to 498-530 nm. The tdTomato fluorophore was excited detected with a 561 nm laser and detected via photomultiplier tubes with a collection window set to 571−610 nm.

**Electron microscopy.** Zebrafish embryos or tissue sections from adult animals were fixed in a 1.5% glutaraldehyde, 3% formaldehyde, 0.1 M cacodylate solution for at least 1 h at room temperature and rinsed in 0.1 M cacodylate buffer three times for 10 min each. Larvae were then stained in 1% $OsO_4$ in veronal acetate for 1 h followed by a 10-minute rinse in veronal acetate. Larvae were then treated with a 1% tannic acid solution in 0.1 M cacodylate solution twice for 20 min each, and stained with 0.5% Uranyl Acetate in veronal acetate at 4 °C overnight. Larvae were rinsed in water twice for 10 min each, embedded in

agarose at 55 °C, and taken through a dehydration series in ethanol (35%, 50%, 75%, 95%, 15 min each), followed by four 10-minute washes in 100% ethanol and 4 10-minute washes in propylene oxide, and a 1 h incubation in a 1:1 mixture of propylene oxide and resin (Epon+Quetol (2:1)+Spurr (3:1) + 2%BDMA) followed by an overnight evaporation. Three additional 1-hour washes were carried out in 100% resin, followed by a final embedding in 100% resin at 55 °C overnight and 70 °C for 4 days. Samples were sectioned on a Reichert Ultracut-S (Leica Microsystems), and imaged with a Phillips Technai-12 electron microscope.

**Microsome preparation.** Microsomes were isolated as previously described[9] with slight modification. Briefly, 100 larvae were anesthetized in tricaine and homogenized in 1.5 mL chilled buffer A (250 mM sucrose, 50 mM Tris pH 7.4, 5 mM EDTA, 0.2% Sodium Azide, 1× cOmplete EDTA protease inhibitor cocktail) in a 5 mL Dounce homogenizer. The homogenate was spun at $9500 \times g$ (11,000 rpm) in a chilled benchtop centrifuge to pellet large cellular debris, and the resulting supernatant was adjusted to pH 5.1 with 0.4 N HCl, rocked at 4 C for 10 min, and spun at $20,000 \times g$ (16,000 rpm) for 30 min to pellet microsomes. The resulting pellet was thoroughly resuspended in 1.5 mL buffer B (0.054% Sodium Deoxycholate, 1 mM EGTA, 1 mM MgCl₂, 1 mM Tris pH 7.4) and transferred to a thick-wall polypropylene centrifuge tube (Beckman-Coulter, #349623). Tubes were ultracentrifuged at $250,000 \times g$ (50,000 rpm) at 4 C in a SW55 Ti rotor in a Beckman Optima XL-80K Ultracentrifuge. The resulting supernatant was collected as the soluble ER-lumenal fraction.

**Density gradient ultracentrifugation.** Density gradient ultracentrifugation was carried out as previously described[9] using an Optiprep (D1556, Sigma-Aldrich) gradient optimized to separate lipoprotein subclasses[78]. Briefly, 1 mL samples were supplemented with 500 μL of Optiprep to yield a 20% iodixanol solution and underlayered beneath 1.5 mL layers of 9% and 12% Optiprep/PBS solutions in a 4.9 mL OptiSeal tube (Beckman-Coulter, 362185) using a disposable glass cannula. Tubes were then ultracentrifuged at $327,000 \times g$ (60,000 rpm) for 3 h in a VTi65.2 rotor in a prechilled Beckman Optima XL 80 K Ultracentrifuge at 4 C. Optiseal tubes were then punctured and 14-350 μL were collected via drip elution. The refractive index of each fraction was determined using a Bausch and Lomb refractometer and used to calculate fraction density.

Density profiling of lipoproteins from cultured cells began with induction of VLDL secretion. 1.5 mL of Optimem reduced medium without phenol red (Invitrogen: 11058021) supplemented with oleic acid (120 nM; Sigma-Aldrich) solubilized in fatty acid–free BSA (Sigma-Aldrich) was added to the cells for 16 h. After this, media were centrifuged at low speed to remove cells and cellular debris and the resulting supernatant was used directly for the gradients. For cell extracts, cells were washed and collected by scraping in PBS. The cell pellet was resuspended in 1.5 mL of hypotonic buffer (10 mM Tis-Cl pH 8.8, protease inhibitors), broken with 20 strokes in a cell cracker (EMBL) and incubated for 1 h on ice. Extracts were centrifuged at $16,000 \times g$ (14,000 rpm) for 5 min at 4 °C. The supernatant was adjusted to 50 mm Tris-Cl pH 8.0, 150 mM NaCl and used for gradients. 800 μl of cleared media or cell extracts were mixed with 200 μL of OptiPrep Density Gradient Medium (Stemcell Technologies: 7820). 900 μL of the mixture was transferred to a polycarbonate ultracentrifuge tubes (Beckman-Coulter:343778), overlayered with 100 μL of PBS. Balanced tubes were then loaded into a TLA 120.2 rotor and centrifuged at $472,000 \times g$ (115,000 rpm) in an Optima table-top ultracentrifuge (Beckman-Coulter) for 2.5 h at 16 °C. Following ultracentrifugation, density fractions were collected carefully from the top of the tube into 10 separate fractions of 100 μL each. The refractive index of each fraction was determined using a refractometer and used to calculate solution density. 10 μL of each fraction, and an equivalent amount of media or cell extracts, were immediately mixed with 100ul of PBS and loaded on a Bio-Dot[R] Microfiltration apparatus (Biorad: 170-6545) for transfer on a nitrocellulose membrane. Dried membranes were blocked with 5% milk and incubated overnight with primary antibodies. After incubation with secondary Alexa-conjugated antibodies, the DotBlot was scanned using Odyssey CLx (LI-COR Bioscience) and individual spots were quantified using QuantityOne (Biorad).

The buoyant:dense ratio of APOB in the cell culture medium was calculated by dividing the quantity of APOB in fractions 1 and 2 (buoyant) by the quantity in fractions 8 and 9 (dense).

**Drug treatments.** Chemical inhibitors were suspended in DMSO as 2,000x stocks and stored at −20 °C: Brefeldin-A (20 mg/mL, Cayman Chemical, #11861), Nocodazole (10 mg/mL, Cayman Chemical, #13857), and Thapsigargin (2 mM, Cayman Chemical, #10522). Working stocks were prepared immediately prior to exposure by supplementing 50 mL of embryo medium with 25 μL of 2000× stock and 25 μL of DMSO. For dual-drug treatments, the DMSO was replaced with 25 μL of 2000× stock of the second drug to maintain a constant concentration of 0.1% DMSO. Larvae were incubated in 1× working stock of Thapsigargin and/or Nocodazole for 90 min, and then anesthetized in tricaine and transferred to homogenization buffer for microsome isolation. Brefeldin-A treatments were performed for 12 h, from 3.5 to 4 dpf, and microsomal contents were isolated as described above.

**Western blotting.** Media and cell extracts prepared as described above were mixed with Laemmli SDS sample buffer and denatured for 10 min at 37 °C, and subjected to SDS-PAGE (6% or 12% acrylamide) followed by Western blotting with the relevant primary antibodies. Following incubation with either Alexa-conjugated or HRP-conjugated secondary antibodies, the blot were imaged using an Odyssey Clx (LI-COR Biosciences) or an Amersham Imager 600, respectively. QuantityOne (Biorad) was used for quantification of individual bands.

The NZYColour Protein Marker II (NZYtech, #MB09003) or Spectra Multicolor High Range Protein Ladder (Thermofisher, #26625) were used as molecular weight standards. The antibodies used for western blotting are as follows: rabbit anti-Antitrypsin Ab-1 (Neomarkers #RB-367-A, RRID:AB_59584, 1:2000 Dilution), Goat anti-Apolipoprotein-B (TEBU-BIO #600-101-111, RRID:AB_2056958, 1:5000 Dilution), Rabbit anti-Calnexin (Abcam #ab22595, RRID:AB_2069006, 1:3000 Dilution), Rabbit anti-collagen XII (Santa Cruz #sc-68862, RRID:AB_2081581, 1:1000 Dilution), mouse anti-MTP C-1 (Santacruz #sc-515742, 1:1000 Dilution), Mouse anti-PDI C-2 (Santacruz #sc-74551, 1:1000 Dilution), Rabbit anti-PLA2G12B (Novus Biologicals #NBP231685, 1:1000 Dilution), Mouse anti-FLAG M2 (Sigma #F1804, RRID:AB_262044, 1:2000 Dilution), Mouse anti-beta actin clone AC-15 (Sigma #A1978, RRID:AB_476692, 1:2000 Dilution). The antibodies used for immunofluorescence were as follows: Goat anti-GRASP65 C-20 (Santa Cruz #sc-19481, RRID:AB_2232631, 1:500 Dilution), and Mouse anti-Sec31A (BD Biosciences #BD-612350, 1:500 Dilution).

**Triglyceride transfer assays.** Triglyceride transfer activity was measured in vitro using an established method[9] scaled down to 384-well plate format. Donor and acceptor vesicles were prepared following established protocols[9,79]. Briefly, 450 nmol of phosphatidylcholine (PC) and 14 nmol of nitrobenzoxadiazole (NBD)-labeled triglyceride (TAG) per milliliter were used in the preparation of donor vesicles (DVs) while in acceptor vesicles (AVs), 2,400 nmol PC/ml was used. NBD-labeled TAG (#6285) and egg PC (#131601 C) were purchased from Setareh Biotech and Avanti Polar Lipids, respectively. For the preparation of DVs and AVs, the desired amount of appropriate lipids in chloroform was added to a glass vial, and lipids were completely dried under the flow of nitrogen. To each dried lipid sample, the required volume of the vesicle buffer (15 mM Tris-HCl, pH 7.4, 40 mM

NaCl, 1 mM EDTA, and 0.02% NaN3) was added, and lipids were mixed and dissolved properly by vortexing. The lipids in the glass vials were sonicated on ice for 20–40 min until the solution became clear. For sonication, a Fisher Scientific 550 Sonic Dismembrator was used and lipid samples were monitored intermittently to avoid heating of the lipid samples. To remove invisible particles, each lipid samples were centrifuged at 4 °C for 1 h using a SW55 Ti rotor at 250,000 × $g$ (50,000 rpm) which was followed by storing clear supernatants in new glass vials at 4 °C for further use.

Each reaction contained 4 µL of donor particles, 4 µL of acceptor particles, 4 µL of mouse microsomal extracts normalized to 400 µg/mL total protein (unless otherwise noted), 4 µL of 10 mM Tris-HCl pH 7.4, 4 µL of appropriate salt solution (maximum 200 mM CaCl₂), and 20 µL of water for a final reaction volume of 40 µL. Microsomal extracts were added last, and after brief mixing by aspiration, the plate was transferred to a pre-warmed plate reader set to 37 °C and fluorescence was quantified every minute for 2 h at 460 nm excitation and 530 nm emission. VLDL from human plasma (Lee Biosciences, #365-10) were diluted 50-fold in water and 4 µL of diluted lipoproteins were used in place of standard acceptor vesicles to monitor transfer to lipoprotein particles.

**Metabolic profiling.** Unbiased metabolic profiling was performed using dual energy X-ray absorptiometry (DEXA) scans and comprehensive lab animal monitor systems (CLAMS). Echocardiograms were performed on conscious mice at baseline and 5 and 10-minute time points following dobutamine injection. Dobutamine was injected IP at a dose of 1.5ug/g body weight. For fractionation of lipoproteins with FPLC, mice were fasted for 4 h prior to sacrifice, and whole blood was collected in EDTA-coated tubes.

**Hypercholesterolemia and atherosclerosis induction in mice.** Adeno-associated virus constructs (AAV8-D377Y-mPCSK9) were procured from the Vector Biolabs 293 Great Valley Parkway Malvern, PA, USA. These viral vector constructs were injected intravenously (IV) in Wild-type (WT) and *PLA2G12B^hlb218* mutant mice at the concentration of 10¹¹ vector genome copies. Thereafter, mice were fed with a Western diet (Envigo, catalog number TD.88137, 21.2% fat and 1.5 g cholesterol per kg) for the period of the experiment. At the end of the experiment, mice were fasted and euthanized followed by plasma collection and evaluation of atherosclerosis. Plasma was collected to document changes in the plasma triglyceride and cholesterol levels. Size exclusion chromatographic separation of plasma lipoproteins was performed on AKTA pure fast performance liquid chromatography (FPLC). Briefly, 300 µL plasma from WT and KO mice after 6 months of WD feeding were analyzed by FPLC column [Superose™ 6 Increase 10/300 GL FPLC column (GE Healthcare)]. Plasma samples were loaded and then eluted with PBS at a flow rate of 0.4 mL/minute and 0.25 mL fractions were collected. Plasma cholesterol and triglyceride levels were measured using kits (Thermo Fisher Scientific, #TR22421 and #TR13421) according to the manufacturer's protocol. Effect of the Pla2g12b knockout on the cholesterol and triglyceride content in very-low-density lipoprotein (VLDL), Intermediate-density lipoprotein (IDL)/low-density lipoprotein (LDL), and high-density lipoprotein (HDL) fractions has been presented in the Fig. 3. For the evaluation of atherosclerosis, aortas were dissected and examined as per the literature.

**Genotyping.** The *pla2g12b^sa659* allele was genotyped by using a forward primer with the sequence ACAAGGGAAAGCAAACCAAA and the reverse primer CAGTGTTGTACATGGTGTCTGC to amplify the polymorphic region (57 °C Tₐ, 30″ extension time). The product was subsequently digested with the restriction enzyme BtsI-v2 (NEB, #R0667S), which has a restriction site present only in the sa659 allele. The *PLA2G12B^hlb218* allele was genotyped using the forward primer GGCACACTCTCCTGCTTCAT and reverse primer TCCACGTTGGAGACAAAGCC (56 °C Tₐ, 30″ extension time). Sanger sequencing was then performed to detect the TACCGCT[G/A]TGACGCAAAATT substitution.

**Plasmids and cloning.** The wild-type zebrafish *pla2g12b* expression plasmid was ordered as a gene synthesis from Twist Bioscience (https://www.twistbioscience.com). The Q5 Site-Directed Mutagenesis kit (NEB, #E0554S) was used to introduce variant alleles as described below. All human *PLA2G12B* variant alleles were ordered as gene syntheses from Twist Bioscience (https://www.twistbioscience.com).

**Rescue injections.** Zebrafish embryos were injected at the 1-cell stage with 30 pg of rescue plasmid and 60 pg of capped Tol2 transposase mRNA in an injection mix containing 1 U/µL SUPERase In RNase inhibitor (ThermoFisher Scientific, AM2694) and 0.05% Phenol Red. At 2 dpf, morphologically normal larvae were visually screened for GFP expression in the yolk using a Zeiss Axiozoom v16 microscope and GFP-negative larvae were discarded. GFP-positive larvae were subsequently anesthetized and imaged as described above and manually scored for rescue activity. Rescue activity was quantified by estimating the percentage of the remaining yolk surface area displaying the translucent coloration characteristic of the wild-type yolk in 10% increments. Representative images and their corresponding scores are provided in Supplementary Fig. 6.

**Co-immunoprecipitation and mass spectrometry.** Co-immunoprecipitation was performed using the FLAG immunoprecipitation kit (Millipore Sigma, #FLAGIPT1-1KT) following manufacturer instructions. Prior to processing, *pla2g12^sa659* larvae were injected with a rescue plasmid containing the *pla2g12b* allele of interest following the rescue injection protocol described above. Injections of ~300 larvae were generally sufficient to yield at least 50 GFP-positive embryos, which were crosslinked in 2% PFA for 20 min at room temperature at 3 dpf. Larvae were homogenized in batches of 10 using pellet pestles in 100 µL of lysis buffer, and the five batches were then pooled, and the total volume was adjusted to 1 mL. Samples were rocked at 4 °C for 30 min and ultracentrifuged in a TLA-100 rotor at 74,000 × $g$ (44,000 rpm) for 20 min to pellet large aggregates. The resulting supernatant was pre-cleared with protein-A coated beads for 2 h at 4 °C, and finally probed with anti-FLAG M2-coated beads overnight at 4 °C. Beads were submitted to the Mass Spectrometry Proteomics Core at Baylor College of Medicine for protein identification on a Thermo Scientific Elite mass spectrometer using published methods[80].

The affinity-purified protein and its interacting proteins were digested on beads. The beads were washed with cold PBS twice and resuspended with 50 µl of 20 mM Ammonium bicarbonate (pH 8.0), 2 mM CaCl₂, and 500 ng of MS-grade trypsin was added and digested overnight at 37 °C. The digestion was stopped by adding 5 µl of 10% formic acid. The digested peptide was removed and enriched by in-house STAGE tip[81] column with 2 mg of C18 beads (3 µm, Dr. Maisch GmbH, Germany) then vacuum dried. Resuspended peptides were subjected to a nanoLC-1000 (Thermo Scientific) coupled to Orbitrap Fusion Lumos™ mass spectrometer (Thermo Scientific) with ESI source. The peptides were loaded onto an in-house Reprosil-Pur Basic C18 (1.9 µm, Dr. Maisch GmbH, Germany) trap column (2 cm length, 100 µm i.d.) and separated by 5 cm column (150 µm i.d.) with a 75 min gradient of 2–28% of acetonitrile/0.1% formic acid at a flow rate of 800 nl/min. The data acquisition was made in data-dependent analysis mode (DDA) for unbiased peptide detection. The precursor MS spectrum was scanned at 300–1400 m/z, 120,000 resolution at 400 m/z, 5 × 10⁵ AGC target (50 ms maximum injection time) by Orbitrap. Top 30 scan was applied to selected MS1 signal and filtered by Quadrupole (2 m/z isolation window, 15 s exclusion time with ±7 ppm mass tolerance), fragmented by Higher-energy C-trap dissociation (HCD), and

detected by Ion trap with rapid scan rate ($5 \times 10^3$ AGC target, and 35 msec of maximum injection time). Obtained spectra were searched against the target-decoy mouse RefSeq database (release Dec. 2020, 28,456 protein sequence) in Proteome Discoverer 2.1 interface (PD 2.1, Thermo Fisher) with the Mascot algorithm (Mascot 2.4, Matrix Science). Dynamic modifications of the acetylation of protein N-terminus and oxidation of methionine were allowed. The precursor mass tolerance was confined within 20 ppm with fragment mass tolerance of 0.5 Da and a maximum of two missed cleavages was allowed. Assigned peptides were filtered with 1% false discovery rate (FDR) using percolator validation based on $q$ value. Calculated area under curve of peptides was used to calculate iBAQ for protein abundance using in-housed software[82].

**Generation of PLA2G12B KnockOut cell lines.** To generate Caco2 and HepG2 cell lines in which PLA2G12B expression was abolished, we performed genome editing with the CRISPR/Cas9 system. The online CRISPR Design Tool (https://chopchop.cbu.uib.no/) was used to select two 20-nt guide sequence for PLA2G12B gene targeting: sgRNA60 5′ GCCAGTACCTACCTGGCTGC3′ and sgRNA94 5′AGGAAGAAG TGATTCCTTCC3′. Construction of the expression plasmid for sgRNA involved a single cloning step with a pair of partially complementary oligonucleotides for each sgRNA. The oligo pairs encoding the 20-nt guide sequences (sgRNA60F 5′CACCGGCCAGTACCTACCTGGCTGC3′ and sgRNA60R 5′AAACGCAGCCAGGTAGGTACTGGCC3′; sgRNA95F 5′ CACCGAGGAAGAAGTGATTCCTTCC3′ and sgRNA94R 5′AAACG GAAGGAATCACTTCTTCCTC3′) were annealed and ligated into the plasmid PX458 using the Bbs I cloning site[83]. The bicistronic pSpCas9 (BB)-2A-GFP (PX458) vector containing cDNAs encoding human codon-optimized Streptococcus pyogenes Cas9 (hSpCas9) with 2A-EGFP, and the remainder of the sgRNA as an invariant scaffold immediately following the oligo cloning site Bbs I, is available from Addgene (plasmid # 48138). Both Caco2 and HepG2 cells were transfected with 1 μg of pooled plasmid using X-tremeGENE9 Transfection Reagent (Roche:6365779001) following manufacturer recommendations. For both cell lines, 96 h after transfection, EGFP-positive cells were isolated by FACS using the FACSAria III from BD Biosciences and FACS Diva 8.0.2 software, and single cells were collected in 96-well plates. After expansion to six-well format, differentiated Caco2 and HepG2 cells were collected, and protein lysates were prepared to identify lines lacking detectable PLA2G12B protein.

**Design of pla2g12b alleles via paralog recoding.** To identify a subset of residues most likely responsible for its function in TRL expansion, we searched for residues that were conserved in *pla2g12b* orthologs from various species that differed from the catalytically active ohnolog *pla2g12a*. Protein sequences were downloaded from Ensembl (ensembl.org) for pla2g12b from zebrafish (ENSDART00000 128494.3, [https://useast.ensembl.org/Danio_rerio/Transcript/Summ ary?db=core;g=ENSDARG00000015662;r=13:4671737-4683149;t=EN SDART00000128494]), mouse (ENSMUST00000009790.14, [https:// useast.ensembl.org/Mus_musculus/Transcript/Summary?db=core;g= ENSMUSG00000009646;r=10:59239482-59257798;t=ENSMUST000 00009790]), and human (ENST00000373032.4, [https://useast. ensembl.org/Homo_sapiens/Transcript/Summary?db=core;g=ENSG0 0000138308;r=10:72934762-72954806;t=ENST00000373032]), and pla2g12a from zebrafish (ENSDART00000103406.5, [https://useast. ensembl.org/Danio_rerio/Transcript/Summary?db=core;g=ENSDAR G00000070454;r=1:12061284-12064799;t=ENSDART000001034 06]), mouse (ENSMUST00000029629.15, [https://useast.ensembl. org/Mus_musculus/Transcript/Summary?db=core;g=ENSMUSG0000 0027999;r=3:129672255-129689474;t=ENSMUST00000029629]), and human (ENST00000243501.10, [https://useast.ensembl.org/Homo_ sapiens/Transcript/Summary?db=core;g=ENSG00000123739;r=4: 109709989-109730070;t=ENST00000243501]). Protein sequences

were aligned using the MUSCLE algorithm within the Geneious bioinformatic software. The gapped alignment length was 250 amino acids, of which 68 contained highly conserved or identical residues. As these were conserved between pla2g12b and pla2g12a in multiple species, it was deemed unlikely they are responsible for neofunctionalization of pla2g12b and instead are likely essential to protein folding. The remaining residues in the alignment were manually inspected to identify sites that were identical between all three pla2g12b orthologs, but different from the pla2g12a paralogs.

**Phylogenetic analyses.** Amino acid sequences for PLA2G12A and PLA2G12B were obtained from ensembl.org, aligned using the online MAFFT alignment server (https://mafft.cbrc.jp/alignment/server/ index.html) with standard settings, and trees were generated and visualized using the Phylo server (https://phylo.io).

**Quantification and statistical analysis.** Statistical analyses were performed using Prism software (Version 9.3.1), as well as RStudio (Version 1.4.1717) unless otherwise noted. Details on the statistical tests are described in the associated figure legends. Unless otherwise noted, error bars denote mean ± standard deviation. Geneious (Geneious Prime 2022.2.1) was used for bioinformatic analyses, and FIJI (ImageJ2 V2.3.0/1.53 f) was used for image analysis.

**Reporting summary**
Further information on research design is available in the Nature Portfolio Reporting Summary linked to this article.

## Data availability
All data generated in this study have been deposited in the Harvard Dataverse database under accession code 6DFOSJ (Pla2g12b Drives Expansion of Triglyceride-Rich Lipoproteins). The mass spectrometry data for proteome profiling have been deposited to the Proteome X change Consortium with dataset identifier PXD048516 (Pla2g12b binding proteins in Zebrafish). Source data are provided with this paper.

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

## Acknowledgements

We would like to acknowledge the Mouse Metabolic Phenotyping Core (MMPC) at Vanderbilt University for performing echocardiography, tissue dissections, and FPLC on mouse samples. The Mass Spectrometry Proteomics Core at Baylor College of Medicine provided essential guidance regarding study design and sample preparation for unbiased proteomics and performed mass spectrometry and data analysis, in particular by Dr. Sung Yun Jung. Dr. Daniel Gorelick also provided invaluable resources, infrastructure, and guidance for experiments carried out at Baylor College of Medicine. We would also like to thank Sujith Rajan for preparing the unilamellar donor and acceptor particles used in triglyceride transfer assays, Kirsten Sadler for providing the ER-TdTomato transgenic line, Michael Matunis for his contribution of superannuated ultracentrifuge supplies that were otherwise unattainable due to supply chain disruptions, and Dr. Chen-Ming Fan for

providing critical feedback and writing assistance on the manuscript. This work was financially supported by National Institutes of Health grants R01DK093399 (S.A.F., J.F.R), R01HL158054 (S.A.F. and M.M.H.), F31HL149174 (T.O.C.M.), F31HL139338 (J.H.T.), the Mathers Foundation (S.A.F.), the Carnegie Institution for Science endowment (S.A.F.), Ministerio de Economía y Competitividad (RYC-2016-20919, O.F.), Ministerio de Asuntos Económicos y Transformación Digital, Gobierno de España (PID2019-105518GB-I00, V.M.), and the European Research Council (ERC-2020-SyG-Proposal No. 951146, V.M.).

## Author contributions

Conceptualization—S.F., J.H.T. O.F., V.M., M.H., P.Y.; methodology—J.H.T., O.F., P.Y.; investigation—J.H.T., O.F., P.Y., M.W., T.O.C.M., M.S., M.M., K.M.; resources—E.B.; Writing and Visualization: J.H.T.; review and editing- M.W., O.F., P.Y., T.O.C.M., M.S., E.B., M.M., K.M., J.R., V.M., M.H., S.F.; supervision- J.R., V.M., M.H., S.F.; funding acquisition—J.R., V.M., M.H., S.F.

## Competing interests

The authors declare no competing interests.
