## [Peer Review File · Nature Communications]

Pla2g12b Drives Expansion of Triglyceride-Rich LipoproteinsReviewer #1 (Remarks to the Author):

This is an interesting paper focusing on the role of Pla2g12b for assembly of triglyceride-rich lipoproteins.

1. The novelty of the findings is somewhat limited since hepatic VLDL production has already been shown to be significantly compromised in PLA2G12B-knockout mice. It has also been shown that PLA2G12B-null mice accumulate triglyceride, cholesterol, and fatty acids in the liver and develop severe hepatosteatosis over time even on regular diet (PMID: 27471003). Also PMID: 33042259 linked PLA2G12B to VLDL secretion.

2. It has also been shown that PLA2G12B interacts with MTP (<https://www.genecards.org/cgi-bin/carddisp.pl?gene=PLA2G12B>)

3. The authors state that: Pla2g12b acts immediately downstream of MTP to ensure efficient transfer of triglycerides to secretible nascent TRLs and prevent excessive lipid transfer to unsecretible LLDs, that Pla2g12b participates directly in TRL biogenesis, and may promote efficient channelling of lipids to ApoB by linking it more tightly to the transfer protein Mtp, that Pla2g12b recruits Mtp to the membrane, that PLA2G12B provides calcium to the TRL biogenesis complex to promote efficient lipid transfer to TRLs, and that PLA2G12B transfers lipids from the ER to TRLs (Fig. 6). This reviewer finds it surprising that the protein can have so many functions. Will site-specific mutagenesis affect these different mechanisms differently? PLA2G12B is a secreted protein. How can it remain in the ER to fulfil all these actions?

4. What data supports that Mtp mediates lipid transfer to lipoprotein nascent VLDL particles? Gordon et al (PMID: 8955151) reported that Inhibition of the Mtp blocks the first step of apolipoprotein B lipoprotein assembly but not the addition of bulk core lipids in the second step. Please clarify.

5. It's very interesting that the that CLDs were virtually undetectable in pla2g12b mutants despite clear intracellular lipid accumulation. Can these results be validated in PLA2G12B^{-/-} mice? Likewise, do ER lipid droplets accumulate in these mice?

6. The results indicate that PLA2G12B is important for formation of TG-rich VLDL, and that lack of the protein will result in secretion of TG-poor VLDL. However, in PLA2G12B-deficient cell lines, secretion of the number of particles (apoB) is significantly reduced. This seems to indicate that it is also important for lipidation of earlier steps in VLDL formation and that apoB is sorted to posttranslational degradation in these cell lines. Please clarify.

7. Results indicate that PLA2G12B deficiency leads to reduced proportion of MTP in the ER membrane. Is the cellular content of MTP reduced?

8. Is the lipidome and proteome (lipid and protein composition) altered in lipoproteins secreted from mice lacking PLA2G12B, or is it only the number of triglyceride molecules that is reduced?

9. ApoB secretion should be analysed in primary hepatocytes isolated from the genetically modified mice. Also, is intestinal production of chylomicrons affected?

10. How does calcium influence VLDL assembly; what is the underlying mechanism(s)? This should be clarified.

Reviewer #2 (Remarks to the Author):

In this study, Thierer and co-authors established that PLA2G12B is an important mediator driving the formation and expansion of triglyceride-rich lipoproteins (TRLs). By generating a panel of 12 mutant alleles, the authors eloquently demonstrated the functional activities of PLA2G12B protein domains. They claimed that calcium could enhance lipidation of TRLs and proposed that PLA2G12B

provides calcium to the TRL biogenesis complex. By using zebrafish and murine *in vivo* studies, they clearly showed that PLA2G12B mutant led to reduced secretion of triglyceride-rich lipoproteins.

The research is interesting and novel. Specifically, the results advance the knowledge of intracellular functional activity of PLA2G12B and propose PLA2G12B as a new mediator involving in the packaging of triglyceride-rich lipoproteins within the ER lumen.

I have a few comments as follows:

In Page 5-6 and Figure 4, the authors proved that calcium is essential for TRLs expansion. PLA2G12B mutant in the calcium-binding domain lost this function and the author claimed that PLA2G12B could "provide" calcium to TRL biogenesis complex. However, the data is not sufficient to support this conclusion. Mutants in the calcium-binding domain might lead to impaired function activity of PLA2G12B and the data did not show convincing evidence about calcium supplement function of PLA2G12B protein.

In the study, the authors showed that PLA2G12B recruits MTTP protein to the membrane by co-immunoprecipitation and mass-spectrometry analysis, and western blot analysis of extracted subcellular compartments. It is recommended that a more thorough approach be taken in the mechanistic *in vitro* studies to link PLA2G12B recruitment or interaction with MTTP protein. For example, the proteins interaction analysis can be addressed by protein co-localization by confocal analysis or physical binding measurement by Bio-core.

In Page 6 and Figure 5, the data showed that PLA2G12B *-/-* mice on a high-fat diet and PSK9 overexpression displayed hypolipidemia and resistance to atherosclerosis. How about the liver phenotype of PLA2G12B *-/-* mice since the lipids might be accumulated in hepatocytes? This is important because the authors stated that PLA2G12B might be a valuable new therapeutic target.

Reviewer #3 (Remarks to the Author):

Metabolism of lipids (fatty acids, triglycerides, and cholesterol) play important role in human health/diseases. In vertebrates, lipids are transported in lipoprotein particles of various sizes. Despite extensive characterization of their sizes and contents (lipids & proteins), loading and unloading of molecules into lipoprotein particles is less understood. PLA2G12B was identified as an important player in regulation of lipids and its contents in circulating lipoprotein particles in mice (refs 23-26) and in zebra fish (Ref #22). However, nearly nothing is known about its function. This study is the first one to characterize PLA2G12B function through a series of carefully designed and elegant *in vitro* and *in vivo* (zebra fish and mouse) experiments. The work is novel and relevant. The manuscript well written, clearly explaining the experiments and conclusion. The authors also demonstrate an approach that combines various models to improve understanding of a function of a molecule. Below are some minor suggestions/questions with a few general comments at the very end.

- It would be nice to briefly precise the location of pla2g12bsa659 (when it is first mentioned in the text) and add visualization to the 3D model in Fig. 2d (unless it is the same location as ENU?).

- NanoLuc-ApoB assay shows higher ApoB in crude/serum and lumen but media from CRISPR/cas9 HepG2 and Caco2 shows decreased ApoB. What ApoB antibody was used for the Western blot in cell culture experiments?

- Fig. 1e – it would be nice (at least in the extended data) to see side by side ER-tomato photos of ER in WT and PLA2G12B (instead of just WT nucleus and MUT ER), similar to EM in Fig.1b. n=?

- Although the strategy for selecting residues for site mutagenesis is well explained, the method could be described in a bit more detail in the Method section/Extended data.

Page 6, lines 244 and 248 –precising the use of Western diet (not simply high-fat diet) is important. According to Material and Methods, Western diet was used (include fat and cholesterol content of Western diet in the Methods section).

- Fig.1d – define RLU in the legend
- Legend for Fig1.f – “medium” of cultured human cells?
- Legend of Fig.2b - GFP control? Uninjected?
- Fig. 2c Rescue activity (%) needs to be defined somewhere in the manuscript (Materials or as previously described with Ref). How is ‘reversal of dark-yolk’ measured (pixel color)? How is percentage calculated? Data are mean of $n \geq 2$ but each one has a lot more than 2 measurements (in fact none look like $n=2$)
- Fig.3 a – $n=3$? What are the individual dots? The method section states that 300 larvae were pooled.
- Fig.3 c – Western blots - $n = ?$ Input? Also, it would be nice to see full blots in the extended data with the loading controls visible. Antibody information needs to be added (especially for PLA2G12B) in Methods or Extended data.
- Fig. 3d – Why are the WT biological replicates connected by dashed line to MUT?
- Extended Fig. 8i shows TG and CHOL serum levels in WT and MUT mice. In MUT, the TG is about 20-30 mg/dL (a bit lower than previously published but it could be due to difference in the assay). Looking at the level of TG in MUT in Fig. 5d, the TG concentrations remained under 50 mg/dL suggesting that feeding Western diet affected TG much less than CHOL. How is the TG level nearly 0 mg/dL in 0 weeks post-injection? If anything, it is supposed to increase from basal with PCSK9-AAV?
- TG/CHOL assay for serum (e.g. in Fig.5d)? Same as for plasma in last series of in vivo experiments with PCSK9-AAV?
- Fig. 6 – perhaps, the WT/Normal PLA2G12B would be forming LLD and TLR on the cytoplasmic side instead of the ER lumen?
- Confocal – filters? For ER-tomato?

Some general ideas/discussion for future:

The authors provide some examples of mutation associated in humans with lipids. A more likely application (as suggested by authors) is targeting for therapeutics, especially reabsorption of cholesterol and absorption of lipids in the intestine. Although liver can synthesize and secrete VLDL into circulation, the main source is food and excessive adipose tissue. Also, could be relevant for hepatic lipid storage disorders (not just cardiovascular).

PLA2G12B is categorized as secreted phospholipase (as opposed to cytoplasmic). Since the authors have PLA2G12B antibody, it would be very interesting to see if PLA2G12B is present in serum/circulating lipoprotein particles in WT mice. As the authors mentioned, one possibility is restructuring membranes. Although PLA2G12B is catalytically inactive phospholipase, it is still categorized as phospholipase: it could be part of the secreted lipoprotein particle, embedded into the phospholipid layer together with other lipoproteins (APOB, etc), with its function related to phospholipids rather than triglycerides or cholesterol (p3 line 115 - “ensure efficient transfer of TG to secreted nascent TRLs”). The authors show data consistent with such hypothesis. PLA2G12B mutants are able to secrete lipoprotein particles but they are small/dense. The accumulating micelles are small, highly curved, which is usually achieved by presence of certain phospholipids. Larger lipoprotein particles have less curvature, which might allow for proper stabilization/orientation of proteins necessary for lipidation of the particle. Lastly, it would be very

interesting to see if these mutants are able to absorb dietary fats in the intestine into chylomicrons or if most lipids are simply excreted with the feces and, therefore, liver uses lipid accumulations to increase bile acids production in attempt to increase uptake in the intestine.

Reviewer #4 (Remarks to the Author):

In the manuscript entitled "Pla2g12b Drives Expansion of Triglyceride-Rich Lipoproteins" Thierer and colleagues investigate the process of loading triglycerides onto lipoproteins for transport in serum by examining the optical properties of the yolk syncytial layer of larval zebrafish as an indicator of defects in lipid packaging. They identify an existing zebrafish mutant sa659, with a splice defect in gene pla2g12b, that is characterized by a darkened yolk phenotype. Using comprehensive phenotyping and elegant transgenic rescue approaches in combination with in vitro studies, the cellular localization at the ER membrane and functional protein domains are identified. This process is dependent on calcium binding by PLA2G12B. The importance of PLA2G12B function is further demonstrated in a murine model of atherosclerosis, where Pla2g12b mutant mice are hypolipidemic and protected from cardiac disease. Further, prior GWAS reveal polymorphisms of PLA2G12B promoter linked to plasma triglyceride levels.

Overall, this is a very exciting and comprehensive study that enables new insight into the transport and packaging of triglycerides into lipoproteins. It is of direct clinical and therapeutic relevance as it identifies a novel potential target to modulate serum lipid levels. The study is extremely elegant, starting from a zebrafish mutant, extensive functional characterization, and involving human in vitro and genomic approaches and elegant murine long-term studies. The figures illustrate the enormous breadth of the studies undertaken, and the illustrations help the reader to understand the functional concepts. There are no major criticisms.

Point-by-point response to reviewers:

Reviewer 1:

Reviewer 1 comment 1:

The novelty of the findings is somewhat limited since hepatic VLDL production has already been shown to be significantly compromised in PLA2G12B-knockout mice. It has also been shown that PLA2G12B-null mice accumulate triglyceride, cholesterol, and fatty acids in the liver and develop severe hepatosteatosis over time even on regular diet (PMID: 27471003). Also PMID: 33042259 linked PLA2G12B to VLDL secretion

While we fully acknowledge that knockout studies have been performed previously in mice and zebrafish, we have clarified our introduction to emphasize that PLA2G12B is “*one of a growing list of genes implicated in TRL homeostasis but lacking mechanistic insight*”. We also clearly reference prior work on this topic by stating that “*Although previous studies have suggested that Pla2g12b plays a role in lipid or lipoprotein metabolism^{33,35-39}, it is unclear whether it impacts lipoprotein biogenesis, secretion, intravascular processing, or systemic lipid metabolism*”. Finally, in the discussion we emphasize that the novelty of our findings lies in dissecting the precise molecular role of PLA2G12B in TRL homeostasis by stating that “*Although numerous genes have been identified that affect vertebrate lipoprotein homeostasis by modulating lipid availability⁷², participating in subcellular trafficking^{28,63}, or mediating intravascular processing and turnover of lipoproteins^{73,74}, PLA2G12B represents an entirely new class of proteins that mediates lipid partitioning between LLDs and nascent TRLs*”.

Reviewer 1 comment 2:

**It has also been shown that PLA2G12B interacts with MTTP
(<https://www.genecards.org/cgi-bin/carddisp.pl?gene=PLA2G12B>)**

We were unable to locate the preexisting evidence of MTTP-PLA2G12B protein-protein interactions at the link provided, but would be happy to cite this additional confirmatory evidence if available.

Reviewer 1 comment 3:

The authors state that: Pla2g12b acts immediately downstream of MTP to ensure efficient transfer of triglycerides to secretable nascent TRLs and prevent excessive lipid transfer to unsecretable LLDs, that Pla2g12b participates directly in TRL biogenesis, and may promote efficient channelling of lipids to ApoB by linking it more tightly to the transfer protein Mtp, that Pla2g12b recruits Mtp to the membrane, that PLA2G12B provides calcium to the TRL biogenesis complex to promote efficient lipid transfer to TRLs, and that PLA2G12B transfers lipids from the ER to TRLs (Fig. 6). This reviewer finds it surprising that the protein can have so many functions. Will site-specific mutagenesis affect these different mechanisms differently? PLA2G12B is a secreted protein. How can it remain in the ER to fulfil all these actions?

We apologize for the confusion regarding the specific activity of PLA2G12B, which we believe stems from inclusion of overly speculative discussion points throughout the manuscript. We have incorporated additional data to strengthen the central conclusions of the manuscript, and refined or removed overly speculative points as follows:

-One of the key conclusions of our paper is that “*although PLA2G12B is a deeply conserved member of the secreted phospholipase gene family, it does not act as a phospholipase, nor is it secreted*”. Although the previous version of the manuscript had some data to support this conclusion, we have incorporated additional data and expanded discussion of this key point.

- We have added *in vivo* imaging experiments of mScarlet-PLA2G12B fusion proteins and referenced human proteomic data that demonstrate PLA2G12B is not secreted: “Microinjection of similar mScarlet-Pla2g12b expression constructs into zebrafish larvae under control of a YSL-specific promoter revealed that expression was restricted to the YSL, with no evidence of secretion into the bloodstream (Fig. 3B), which is consistent with previous analyses of the human serum proteome showing that PLA2G12B is virtually undetectable in the human bloodstream⁴⁹.”
- We have incorporated additional western blots demonstrating that PLA2G12B is undetectable in the cell culture media of both Caco2 (intestinal) and HepG2 (Liver) cell lines: “*Further, western blots demonstrated that PLA2G12B is undetectable in the media (Fig. S4E,F, S9). These findings demonstrate that PLA2G12B is not secreted, but instead is retained within the ER.*”

-The statement “*that Pla2g12b participates directly in TRL biogenesis*” was overly vague and has been removed from the manuscript.

-PLA2G12B contains a calcium-binding domain that is essential for protein function. The statement that PLA2G12B may “*provide calcium to the TRL biogenesis complex*” was overly speculative, and has been removed from the manuscript. Instead, we provided a more grounded discussion where we highlight our novel finding that calcium selectively enhances lipidation of APOB while emphasizing that the precise mechanistic role of calcium remains unclear: “*Although we initially investigated the role of calcium in TRL biogenesis because of the calcium-binding motif present in PLA2G12B, our in vitro assays demonstrated that calcium greatly enhances lipidation of APOB in the absence of PLA2G12B (Fig. 4). As the lipid transfer activity of MTP is unaffected by calcium supplementation when APOB-free lipid micelles are used as acceptor vesicles, the most parsimonious interpretation is that calcium interacts with the APOB and enhances its ability to serve as a lipid acceptor. This hypothesis is in line with previous reports that APOB binds calcium⁶⁶, and that MTP transfer to TRLs can be activated by calcium⁶⁷. However, the precise role of the calcium-binding domain of PLA2G12B, and the broader role of calcium in TRL expansion warrant further investigation.*”

-One of the key mechanistic findings of this paper is “*that PLA2G12B increases the proportion of MTP bound to the ER-membrane (Fig. 3G,H)*”. This observation, along with the enhanced lipidation of APOB in the presence of PLA2G12B (Fig. 1E,F), and previous reports that “*the initial stages of APOB lipidation occur along the ER membrane⁷⁰*”, led us to forward the following hypothesis in the discussion section: “*it is possible that concentration of MTP along this two-dimensional surface contributes to enhanced lipidation of APOB*”. This statement is followed by discussion of additional alternative hypotheses, which propose that “*membrane remodeling*” or “*fusion between TRLs and LLDs*” may be responsible for the enhanced lipidation of TRLs in the presence of PLA2G12B. We believe that this language clearly differentiates between the conclusions that are directly supported by the data and the more speculative discussion points that integrate prior literature into our proposed model. Further, we have modified our model to utilize solid arrows to indicate findings directly supported by the data, and dashed arrows to

highlight areas where uncertainty remains regarding the precise transfer routes for lipids within the ER membrane (Fig. 6).

Reviewer 1 Comment 4:

What data supports that Mtp mediates lipid transfer to lipoprotein nascent VLDL particles? Gordon et al (PMID: 8955151) reported that Inhibition of the Mtp blocks the first step of apolipoprotein B lipoprotein assembly but not the addition of bulk core lipids in the second step. Please clarify.

We have greatly expanded discussion of this topic, as it represents a persistent knowledge gap in TRL assembly. In the introduction, we point out that “*TRL expansion appears to occur in two phases^{17,20,21}. The first phase involves direct transfer of a small amount of lipids to APOB by MTP, forming a small nascent particle with densities similar to high-density lipoprotein. Subsequently, a second phase of lipidation occurs whereby a large amount of lipids (predominantly triglycerides) are rapidly added to the particle core, creating a mature TRL. This second phase of lipidation is not blocked by inhibitors of MTP lipid transfer activity¹⁷, suggesting it proceeds through a distinct mechanism than the first phase⁵. As mentioned above, it has been speculated that fusion between small TRLs and LLDs could explain the rapid expansion of TRLs in this second phase of lipidation, but fusion-independent models have not been conclusively ruled out*”.

We have incorporated a reference to the paper cited above that uses a photoactivatable MTP inhibitor to demonstrate that the second stage of TRL expansion can proceed despite transient inhibition of MTP transfer activity. We have also added a reference that identifies fusogenic activity of MTP peptides, which would represent a transfer-independent mechanism by which MTP could incorporate lipids into nascent TRLs. MTP is unequivocally required for the initial lipidation of TRLs, and for production of LLDs. As fusion with LLDs is one potential mechanism by which lipids are transferred to TRLs in the second lipidation/expansion step, MTP is still indirectly required to generate the LLDs involved in the second TRL lipidation/expansion step even if it is not directly involved in the fusion reaction. We believe the language above acknowledges that although transient inhibition of MTP lipid transfer activity does not block the second phase of TRL expansion, this observation alone does not exclude the possibility that the fusogenic or the LLD-generating activities of MTP could still be required for the second phase of lipidation. That being said, in response to this concern we have revised our model to account for potential MTP-independent pathways for particle expansion (Fig. 6), such as fusion events between LLDs or phospholipid membrane remodeling.

Reviewer 1 Comment 5:

It's very interesting that the that CLDs were virtually undetectable in pla2g12b mutants despite clear intracellular lipid accumulation. Can these results be validated in PLA2G12B-/- mice? Likewise, do ER lipid droplets accumulate in these mice?

We apologize for the insufficient discussion of this phenomenon in the initial submission, and have added a new section to the discussion dedicated to this topic as follows:

“Knockout studies in mice have previously reported that cytoplasmic lipid droplets accumulate in lipoprotein-producing tissues of PLA2G12B mutants. This contrasts sharply with

our findings in the YSL of PLA2G12B mutant zebrafish, which are devoid of CLDs and instead accumulate LLDs. Consistent with previous studies in mice, we found that adult tissues in zebrafish also accumulate CLDs rather than LLDs (Fig. S12D,E). We therefore speculate that differences in the stage of development, rate of TRL production, and levels of luminal lipase expression (such as Triacylglycerol hydrolase, TGH/Ces3)^{18,68} may drive differences in CLD and LLD lipid partitioning between the YSL and adult tissues”.

Reviewer 1 Comment 6:

The results indicate that PLA2G12B is important for formation of TG-rich VLDL, and that lack of the protein will result in secretion of TG-poor VLDL. However, in PLA2G12B-deficient cell lines, secretion of the number of particles (apoB) is significantly reduced. This seems to indicate that it is also important for lipidation of earlier steps in VLDL formation and that apoB is sorted to posttranslational degradation in these cell lines. Please clarify.

We apologize for the insufficient discussion of this phenomenon in the initial submission, and have added a new section to the discussion dedicated to this topic as follows:

“We detect reduced secretion of APOB in all model systems tested, and a transient increase in luminal APOB levels in pla2g12b mutant larval zebrafish. However, secretion of other factors, including large cargoes such as collagen that require specialized secretory machinery, remains intact⁴⁴. Further, we found that pharmacological inhibition of the secretory pathway with BFA increased levels of luminal APOB in PLA2G12B mutants. These findings suggest that although APOB secretion is lower in PLA2G12B mutants, PLA2G12B is not required for TRL secretion. Instead, our results suggest that the defects in TRL secretion may be a secondary consequence of improper TRL expansion, which could compromise interactions with sorting receptors required for TRL secretion⁶³ or activate quality control machinery that degrades defective TRLs^{4,8,64}”

Reviewer 1 Comment 7:

Results indicate that PLA2G12B deficiency leads to reduced proportion of MTP in the ER membrane. Is the cellular content of MTP reduced?

We have added a supplemental figure to address this question, Fig. S9, and added a reference to this figure in the main text:

“We found that although the cellular MTP content remains unchanged (Fig. S9), a significantly lower proportion of MTP was associated with the membrane in PLA2G12B^{-/-} mutant HepG2 cells (Fig. 3G,H).”

Reviewer 1 Comment 8:

8. Is the lipidome and proteome (lipid and protein composition) altered in lipoproteins secreted from mice lacking PLA2G12B, or is it only the number of triglyceride molecules that is reduced?

We have expanded discussion of this topic by moving the FPLC traces of triglyceride and cholesterol levels from the supplemental text to main figures (Fig. 5A-H), and independently quantified the levels of these lipids present in each lipoprotein class. While comprehensive analyses of the lipidome and proteome of these particles is outside of the scope of this work, we

do conclusively demonstrate a significant depletion of triglyceride levels in the VLDL fraction despite surprisingly unperturbed levels of VLDL cholesterol (Fig. 5B,F) as described below:

“To determine the precise changes in lipoprotein composition contributing to hypolipidemia, we performed FPLC and enzymatic lipid profiling on plasma isolated from PLA2G12B^{hb218} mutant mice and their wild-type siblings (Fig. 5A-H). PLA2G12B^{hb218} is a point mutation generated by ENU that substitutes a cysteine residue at position 129 with tyrosine (C129Y)³⁷. As expected, we found that very-low density lipoproteins (VLDL, the triglyceride-rich lipoproteins secreted from the liver) exhibited a >80% reduction in triglyceride content (Fig. 5B). However, despite this drastic decrease in triglyceride content, the cholesterol content of VLDL particles remained unchanged (black arrowhead Fig. 5E,F). We conclude that the abnormal TRLs secreted by Pla12g12b mutants are selectively depleted in triglycerides.”

Reviewer 1 Comment 9:

ApoB secretion should be analysed in primary hepatocytes isolated from the genetically modified mice. Also, is intestinal production of chylomicrons affected?

While we concur that studies in primary hepatocytes would have value, we decided to prioritize studies in human cells, which would have the highest translational relevance. This involved generating multiple new pla2g12b mutant cell lines (HepG2 and Caco2, Figs. 2, 3, S4). These studies in human intestinal cell lines also allow us to extend our findings by supporting the conclusion that Pla2g12b has a role in chylomicron biogenesis.

Reviewer 1 Comment 10:

How does calcium influence VLDL assembly; what is the underlying mechanism(s)? This should be clarified.

We have modified the discussion and softened our conclusions to emphasize that although we identify calcium as an essential and previously unappreciated cofactor for TRL maturation, the mechanistic role of calcium remains unknown, and will be the focus of future work as described below:

“Although we initially investigated the role of calcium in TRL biogenesis because of the calcium-binding motif present in PLA2G12B, our in vitro assays demonstrated that calcium greatly enhances lipidation of APOB in the absence of PLA2G12B (Fig. 4). As the lipid transfer activity of MTP is unaffected by calcium supplementation when APOB-free lipid micelles are used as acceptor vesicles, the most parsimonious interpretation is that calcium interacts with the APOB and enhances its ability to serve as a lipid acceptor. This hypothesis is in line with previous reports that APOB binds calcium⁶⁶, and that MTP transfer to TRLs can be activated by calcium⁶⁷. However, the precise role of the calcium-binding domain of PLA2G12B, and the broader role of calcium in TRL expansion warrant further investigation”.

Reviewer 2:

Reviewer 2 Comment 1:

In Page 5-6 and Figure 4, the authors proved that calcium is essential for TRLs expansion. PLA2G12B mutant in the calcium-binding domain lost this function and the

author claimed that PLA2G12B could “provide” calcium to TRL biogenesis complex. However, the data is not sufficient to support this conclusion. Mutants in the calcium-binding domain might lead to impaired function activity of PLA2G12B and the data did not show convincing evidence about calcium supplement function of PLA2G12B protein.

Reviewer 1 also shared this concern, and we agree that this statement was overly speculative and have removed it from the manuscript and refined discussion as outlined in the response to Reviewer 1 comment 3 (copied below for convenience)

The statement that PLA2G12B may “provide calcium to the TRL biogenesis complex” was overly speculative, and has been removed from the manuscript. Instead, we provided a more grounded discussion where we highlight the novel finding “*that calcium greatly enhances lipidation of APOB in the absence of PLA2G12B*”, and emphasize that “*the precise role of the calcium-binding domain of PLA2G12B, and the broader role of calcium in TRL expansion warrant further investigation*”.

Reviewer 2 Comment 2:

In the study, the authors showed that PLA2G12B recruits MTTP protein to the membrane by co-immunoprecipitation and mass-spectrometry analysis, and western blot analysis of extracted subcellular compartments. It is recommended that a more thorough approach be taken in the mechanistic in vitro studies to link PLA2G12B recruitment or interaction with MTTP protein. For example, the proteins interaction analysis can be addressed by protein co-localization by confocal analysis or physical binding measurement by Bio-core.

We have incorporated additional data, softened discussion of these interactions in the introduction, results, and discussion, and provided a reviewer-only figure to address this concern as follows:

-Additional experiments have been added using confocal imaging to localize Pla2g12b within the ER:

“Although PLA2G12B is a member of a secreted family of phospholipases, the presence of an amino-terminal signal peptide and carboxy-terminal ER-retention motif (KDEL) suggest that this protein would be retained within the ER. We designed additional alleles of PLA2G12B fused to mScarlet to visualize PLA2G12B localization in vivo. HepG2 cells were transfected with chemically inducible mScarlet-Pla2g12b expression constructs and co-stained with ER-tracker dye, revealing clear co-localization with the ER (Fig. 3A). Sustained expression of this construct under control of a constitutive promoter led to formation of fibrous intracellular aggregates, which is likely an artifact of overexpression (Fig. S8A). Microinjection of similar mScarlet-Pla2g12b expression constructs into zebrafish larvae under control of a YSL-specific promoter revealed that expression was restricted to the YSL, with no evidence of secretion into the bloodstream (Fig. 3B), which is consistent with previous analyses of the human serum proteome showing that PLA2G12B is virtually undetectable in the human bloodstream⁴⁹. Importantly, the mScarlet-Pla2g12b fusion construct reversed the darkened-yolk phenotype in a dose dependent manner (Fig. 3C), and also displayed a reticular pattern under confocal imaging (Fig. S8B). Further, western blots demonstrated that PLA2G12B is undetectable in the media (Fig. S4E,F, S9). These findings demonstrate that PLA2G12B is not secreted, but instead is retained within the ER.”

-Language supporting conclusive MTP-PLA2G12B interactions has been removed from the Abstract and Introduction

-Language has been softened in the results section to reflect the possibility that co-immunoprecipitated proteins may not necessarily reflect direct binding partners:

“We conclude that PLA2G12B is retained within the cell where it associates tightly with the ER membrane as well as numerous key players in TRL biogenesis and concentrates the lipid transfer protein MTP along the luminal face of the ER membrane. Unfortunately, we were not able to successfully secrete and purify recombinant PLA2G12B (data not shown), precluding further validation of protein-protein interactions in vitro. As the ER membrane is a site of active TRL biogenesis, particularly within the YSL, additional studies will be required to distinguish direct protein-protein interactions from incidental interactions resulting from colocalization along the ER membrane.”

-Language has also been softened in the discussion section to reflect the possibility that co-immunoprecipitated proteins may not necessarily reflect direct binding partners:

“These observations demonstrate that the acquisition of triglyceride-transfer activity by MTP was necessary, but not sufficient, to drive the emergence of TRLs in the vertebrate lineage. We identify PLA2G12B as a vertebrate-specific gene that ensures efficient lipidation of TRLs and restricts lipid accumulation in LLDs (Fig. 6). Mechanistically, we find that PLA2G12B increases the proportion of MTP bound to the ER-membrane (Fig. 3G,H). As the initial stages of APOB lipidation occur along the ER membrane⁷, it is possible that concentration of MTP along this two-dimensional surface contributes to enhanced lipidation of APOB. Alternatively, other catalytically inactive phospholipases have been shown to have membranotropic properties⁵⁴, which led us to speculate that membrane remodeling by PLA2G12B could also contribute to enhanced lipidation of TRLs. Finally, several lines of evidence have suggested that TRL expansion occurs through fusion between TRLs and LLDs^{16,17,21,45}. If this fusion model is correct, then it would suggest that PLA2G12B enhances TRL expansion by promoting fusion between TRLs and LLDs. Our results are therefore perfectly consistent with both prevailing models of TRL expansion; PLA2G12B may enhance TRL expansion by promoting transfer of lipids from the ER membrane directly to TRLs, or lipids may be transferred to an LLD intermediate prior to being incorporated into TRLs (Fig. 6). “

Finally, although we were initially excited by the prospect of *in vitro* interaction studies (such as Biacore SPR), we were unable to purify appreciable quantities of protein. See reviewer-only figure below, demonstrating significant retention of recombinant PLA2G12B within the cell despite addition of a secretion tag. We are therefore skeptical that recombinant protein will be functional in the soluble fraction, as proper folding likely requires interactions with lipid membranes.

Purification of PLA2G12B:

Reviewer 2 Comment 3:

In Page 6 and Figure 5, the data showed that PLA2G12B $-/-$ mice on a high-fat diet and PSK9 overexpression displayed hypolipidemia and resistance to atherosclerosis. How about the liver phenotype of PLA2G12B $-/-$ mice since the lipids might be accumulated in hepatocytes? This is important because the authors stated that PLA2G12B might be a valuable new therapeutic target.

We have expanded discussion on this topic to emphasize the tradeoffs associated with using PLA2G12B as a potential therapeutic target as follows:

“It has been proposed that selectively interfering with the triglyceride-transfer activity of MTP may provide therapeutic value by allowing essential lipids to flow between tissues in triglyceride-poor particles, which may be less likely to induce serum hyperlipidemia and cardiovascular disease^{12,69}. Our assessment of cardiovascular disease in PLA2G12B mutants provide the first experimental evidence that interfering with TRL expansion is atheroprotective. However, the diminished triglyceride content of secreted TRLs leads to lipid retention in lipoprotein-producing tissues like the intestine and liver, suggesting that other dietary or pharmacological interventions may be warranted to limit the effects of steatosis. However, our broad panel of metabolic phenotyping suggested that PLA2G12B mutant animals were remarkably healthy despite evidence of steatosis (Fig. S11), suggesting it may provide an advantageous tradeoff in certain contexts. The human GWAS data indicating that a SNP in an HNF4a transcription factor binding site of the PLA2G12B locus is associated with circulating triglyceride and cholesterol levels in humans suggests that modulating the level of PLA2G12B may be a strategy to produce a less atherogenic plasma lipid profile without the steatosis associated with a null allele.

Modulation of PLA2G12B activity may have particular therapeutic value for the treatment of familial lipase deficiencies, such as lipoprotein lipase (LPL) mutants^{70,71}. These deficiencies do not respond well to standard therapies, which predominantly act by enhancing lipoprotein uptake, but should be treatable through modulation of lipid secretion via PLA2G12B. Specifically, we show that PLA2G12B mutant mice with compromised lipoprotein uptake resulting from PCSK9 overexpression still maintain relatively normal levels of plasma TG (Fig. 5J). While pharmacological inhibition of protein-protein interactions is notoriously difficult, our identification of an essential calcium-binding pocket within PLA2G12B highlights a druggable target domain for rational drug development. PLA2G12B therefore represents a valuable new therapeutic target to combat the growing global prevalence of hyperlipidemia and cardiovascular disease. “

Reviewer 3:

Reviewer 3 Comment 1:

It would be nice to briefly precise the location of pla2g12bsa659 (when it is first mentioned in the text) and add visualization to the 3D model in Fig. 2d (unless it is the same location as ENU?).

We have expanded information on this allele, both when it first appears in the text: “*We identified an existing mutant (sa659) in the zebrafish mutation project collection³³ that exhibits the dark-yolk phenotype (Fig. 1B), and found that the allele disrupts an essential splice site in the poorly characterized gene pla2g12b*”, as well as in the methods section “*The pla2g12b^{sa659} allele was generated via N-ethyl-N-nitrosourea (ENU) mutagenesis³³, and results in substitution of an adenine nucleotide with thymine within the splice acceptor site of intron 3-4 (g.10194A>T). Disruption of this splice acceptor site is predicted to result in failure to incorporate the final exon (exon 4) into the mature transcript and cause nonsense-mediated decay⁷⁵*”, and a deeper discussion of the consequences of this mutation are outlined in the associate reference (Anderson, J. L. *et al.*). As this allele disrupts an essential splice site, its primary mechanism of gene disruption is through nonsense-mediated decay of mRNA rather than alteration of the amino acid sequence. For this reason we have elected not to display the amino acid changes associated with this allele on the 3D model.

Reviewer 3 Comment 2:

- NanoLuc-ApoB assay shows higher ApoB in crude/serum and lumen but media from CRISPR/cas9 HepG2 and Caco2 shows decreased ApoB. What ApoB antibody was used for the Western blot in cell culture experiments?

We have added additional data to address this topic, and clarified and expanded discussion. The NanoLuc-APOB assay in zebrafish shows lower levels of APOB in the serum of *Pla2g12b* mutants: “*Crude lysates (representative of serum lipoproteins) show that pla2g12b mutants secrete a lower number of TRLs overall, and that the secreted particles are significantly smaller and denser than those of wild-type controls (Fig. 1E)*”. Our results from cultured human cells perfectly mirror those from zebrafish: “*Consistent with our results in larval zebrafish studies and previous work in mouse, we do detect decreased overall secretion of APOB in PLA2G12B^{-/-} mutant cell lines (Fig. S4B,D)*”

Further, we show that luminal APOB levels are transiently higher in *pla2g12b* mutant zebrafish: “APOB levels are higher in luminal extracts of *pla2g12b* mutants from 1-3 days post-fertilization (dpf), but drop below wild-type levels by 4 dpf (Fig. 1F)”.

We added additional experiments using trafficking inhibitors to demonstrate that TRL secretion was still intact in *pla2g12b* mutants: “To evaluate whether the elevated levels of APOB in the ER lumen in the early developmental stages resulted from defects in TRL secretion, zebrafish larvae were exposed to the trafficking inhibitor brefeldin-A (BFA). Disruption of the secretory pathway with BFA led to increased accumulation of APOB in the ER lumen in both wild-type and *pla2g12b* mutants (Fig. S3), suggesting that TRL secretion is intact in both genotypes. Consistent with previous reports, we found that BFA also interfered with TRL expansion⁴⁵. We conclude that BFA disrupts both TRL expansion and cellular trafficking, whereas mutations in *pla2g12b* selectively disrupt TRL expansion while leaving TRL secretion intact.”

We have added a complete list of antibodies used in this study as Table S1, which details the anti-APOB antibody used as the Goat anti-Apolipoprotein B from TEBU-BIO, RRID:AB_2056958.

Reviewer 3 Comment 3:

- Fig. 1e – it would be nice (at least in the extended data) to see side by side ER-tomato photos of ER in WT and PLA2G12B (instead of just WT nucleus and MUT ER), similar to EM in Fig.1b. n=?

The initial submission did depict side-by-side comparisons of the ER in WT and Mutant larvae, but may have been unrecognizable as such due to significant distortion of the ER in mutant larvae. To address this, we have incorporated multiple levels of magnification in the figure panels (Fig. 1G) and expanded discussion of our observations as follows:

“We therefore used the fluorescent reporter ER-tdTomato to visualize ER morphology in *pla2g12b* mutants and wild type controls (Fig. 1G). The wild-type ER presented a typical morphology of a network of interconnected tubules with diameters <100 nm. By contrast, the ER was greatly distended in *pla2g12b* mutants, swelling to several microns in diameter and containing large inclusions (white arrowheads, Fig. 1G) resembling those seen in electron micrographs (Fig. 1B).”

The figure legend has been updated to reflect that “Images are representative of n≥4 samples per group.”

Reviewer 3 Comment 4:

- Although the strategy for selecting residues for site mutagenesis is well explained, the method could be described in a bit more detail in the Method section/Extended data.

A new section has been added to the materials and methods section that describes the allele design in greater detail.

“Design of *pla2g12b* alleles via paralog recoding

*To identify a subset of residues most likely responsible for its new function in TRL expansion, we searched for residues that were conserved in *pla2g12b* orthologs from various species that differed from the catalytically active ohnolog *pla2g12a*. Protein sequences were downloaded from*

Ensembl (ensembl.org) for *pla2g12b* from zebrafish (ENSDART00000128494.3), mouse (ENSMUST00000009790.14), and human (ENST00000373032.4), and *pla2g12a* from zebrafish (ENSDART00000103406.5), mouse (ENSMUST00000029629.15), and human (ENST00000243501.10). Protein sequences were aligned using the MUSCLE algorithm within the Geneious bioinformatic software. The gapped alignment length was 250 amino acids, of which 68 contained highly conserved or identical residues. As these were conserved between *pla2g12b* and *pla2g12a* in multiple species, it was deemed unlikely they are responsible for the newly evolved function of *pla2g12b* and instead are likely essential to protein folding. The remaining residues in the alignment were manually inspected to identify sites that were identical between all three *pla2g12b* orthologs, but different from the *pla2g12a* paralogs.”

Reviewer 3 Comment 5:

Page 6, lines 244 and 248 –precising the use of Western diet (not simply high-fat diet) is important. According to Material and Methods, Western diet was used (include fat and cholesterol content of Western diet in the Methods section).

Instances of high-fat diet have been replaced with Western diet, and the catalog number and lipid contents of the diet were added to the methods section as follows:

“Thereafter, mice were fed with a Western diet (Envigo, catalog number TD.88137, 21.2% fat and 1.5 g cholesterol per kg) for the period of experiment.”

Reviewer 3 Comment 6:

- Fig.1d – define RLU in the legend

“ApoB-NanoLuc (measured in relative luminescence units, RLU)”

Reviewer 3 Comment 7:

- Legend for Fig1.f – “medium” of cultured human cells?

“Density gradient profiles of secreted TRLs in the media of cultured human cells”

Reviewer 3 Comment 8:

- Legend of Fig.2b - GFP control? Uninjected?

*“Representative fluorescent and brightfield images of *pla2g12b*^{sa659} mutants injected with rescue plasmid (Injected) and uninjected controls (Uninjected), showing reversal of the dark-yolk phenotype (white arrowhead) and expression of a GFP marker throughout the YSL of successfully injected larvae.”*

Reviewer 3 Comment 9:

- Fig. 2c Rescue activity (%) needs to be defined somewhere in the manuscript (Materials or as previously described with Ref). How is ‘reversal of dark-yolk’ measured (pixel color)? How is percentage calculated? Data are mean of n≥2 but each one has a lot more than 2 measurements (in fact none look like n=2)

We apologize for this omission. We have added a supplemental figure detailing the quantification method (Fig. S6), incorporated a brief description of the method in the main text, and added a more detailed description of the method to the Methods section as follows:

“Rescue activity was quantified by estimating the percentage of the yolk surface area exhibiting the translucent appearance typical of wild-type yolks (Fig. S6). All four alleles that disrupted putative functional domains virtually abolished rescue activity, indicating that these

motifs are all essential for protein function (blue labels, Fig. 2C). Four of the alleles generated through paralog recoding abolished rescue activity as well (red labels, Fig. 2C), which we denote as essential domains A and B.”

“Rescue Injections

Zebrafish embryos were injected at the 1-cell stage with 30 pg of rescue plasmid and 60 pg of capped Tol2 transposase mRNA in an injection mix containing 1 U/ μ L SUPERase In RNase inhibitor (ThermoFisher Scientific, AM2694) and .05% Phenol Red. At 2 dpf, morphologically normal larvae were screened for GFP expression in the yolk, and GFP-negative larvae were discarded. GFP positive larvae were subsequently anesthetized and imaged as described above and manually scored for rescue activity. Rescue activity was quantified by estimating the percentage of the remaining yolk surface area displaying the translucent coloration characteristic of the wild-type yolk in 10% increments. Representative images and their corresponding scores are provided in Fig. S6.”

We apologize for our imprecise reporting of sample size. In this context, $n \geq 2$ was intended to designate that at least two independent trials were performed (different clutches on different days). We have revised our reporting to use n exclusively to reflect the sample size (number of samples or number of animals where appropriate), and provided additional information on the number of trials where appropriate. In this specific case, the text has been modified to:

“Data are mean \pm s.d. of $n \geq 12$ biological replicates pooled from at least two independent experiments”.

Reviewer 3 Comment 10:

- Fig.3 a – $n=3$? What are the individual dots? The method section states that 300 larvae were pooled.

Individual dots represent results from three independent co-IP/MS experiments. The figure legend has been updated to incorporate this information as follows:

“Data are mean \pm s.d. of $n=3$ independent experiments, where each experiment involved pooling of 300 individual larvae to achieve sufficient input protein for co-IP/MS analyses.”

Reviewer 3 Comment 11:

- Fig.3 c – Western blots - $n = ?$ Input? Also, it would be nice to see full blots in the extended data with the loading controls visible. Antibody information needs to be added (especially for PLA2G12B) in Methods or Extended data.

We apologize for these omissions. The figure legend has been updated to reflect the sample number, full blots have been provided in the source data file as well as in an enduring online public repository (See data availability statement, <https://doi.org/10.7910/DVN/6DFOSJ>), and antibody information has been added to the supplementary materials (Table S1).

“(G) Representative western blots of input (I), soluble (S), and membrane (M) cellular fractions of HepG2 cells showing that Pla2g12b partitions exclusively to the membrane fraction, and that less MTP is recruited to the membrane fraction in cells lacking functional PLA2G12B (white arrowhead). Protein disulfide isomerase (PDI) and Calnexin (CNX) are shown as markers of

soluble and input fractions, respectively. **(H)** Quantification of MTP partitioning between soluble and membrane fractions as shown in panel G. Data are from $n=5$ biological replicates. Unpaired two-tailed t -test, $p=.03$. “

Reviewer 3 Comment 12:

Fig. 3d – Why are the WT biological replicates connected by dashed line to MUT?

The dashed lines indicate pairs of samples that were processed in parallel, but have been removed as they do not contribute to interpretation of the results.

Reviewer 3 Comment 13:

Extended Fig. 8i shows TG and CHOL serum levels in WT and MUT mice. In MUT, the TG is about 20-30 md/dL (a bit lower than previously published but it could due to difference in the assay). Looking at the level of TG in MUT in Fig. 5d, the TG concentrations remained under 50 mg/dL suggesting that feeding Western diet affected TG much less than CHOL. How is the TG level nearly 0 mg/dL in 0 weeks post-injection? If anything, it is supposed to increase from basal with PCSK9-AAV?

We have added discussion of this point to the relevant results section:

“Notably, while this hyperlipidemia paradigm caused cholesterol levels to increase dramatically in PLA2G12B mutant mice (>100-fold increase), the increase in serum triglyceride levels was much less pronounced (~7-fold increase).”

And discussed the physiological and therapeutic implications of this finding in the discussion as well:

“Specifically, we show that PLA2G12B mutant mice with compromised lipoprotein uptake resulting from PCSK9 overexpression still maintain relatively normal levels of plasma TG (Fig. 5J).”

We have also clarified the figure legend to convey that the 0 week post-injection time point corresponds to samples collected just prior to injection, which is why plasma lipid levels are low at this time point. Also note that the previously submission only displayed data from male mice, but has been updated to reflect pooled data from male and female samples, and this information has been added to the figure legend:

“(J) Quantification of plasma levels of triglycerides and (K) cholesterol following AAV injection and transition to western diet, where time 0 corresponds to samples collected immediately prior to induction of hyperlipidemia. Data are mean \pm s.d. of $n\geq 5$ gender balanced samples per group.”

Reviewer 3 Comment 14:

TG/CHOL assay for serum (e.g. in Fig.5d)? Same as for plasma in last series of in vivo experiments with PCSK9-AAV?

All assays reflect plasma lipoprotein concentrations, erroneous instances of “serum” have been removed.

Reviewer 3 Comment 15:

Fig. 6 – perhaps, the WT/Normal PLA2G12B would be forming LLD and TLR on the cytoplasmic side instead of the ER lumen?

We have added a cytoplasmic lipid droplet (CLD) to a schematic in Fig. 1A, as well as Fig. 6. CLDs are often decorated by PLIN proteins, but the key distinguishing feature between CLD and LLDs is their subcellular location:

“Although no specific protein marker has been identified for LLDs, they can be identified as the APOB-free subset of lipid micelles within the ER lumen.”

Similarly, TRLs are also exclusively found in the ER lumen, as APOB is degraded unless stabilized by luminal cofactors and lipids. We have added this necessary background to the introduction:

“APOB is translated by ER-bound ribosomes and passes through the translocon⁶. Within the ER, the hydrophobic portions of APOB are stabilized through interactions with the luminal face of the ER-membrane as well as co-translational addition of lipids by MTP⁷. If insufficient lipids are available or MTP activity is compromised, APOB is rapidly degraded via the ER-associated degradation (ERAD) pathway⁸.”

Reviewer 3 Comment 15:

- Confocal – filters? For ER-tomato?

We have added this information to the methods section:

“Confocal images of zebrafish larvae were taken with a Leica DMI6000 inverted microscope using a Leica 63x objective with a Leica TCS-SP5 II confocal scanner. The mCherry fluorophore was excited with a 561nm laser and detected via photomultiplier tubes with a collection window set to 610–650nm.”

Reviewer , miscellaneous comments:

Some general ideas/discussion for future:

The authors provide some examples of mutation associated in humans with lipids. A more likely application (as suggested by authors) is targeting for therapeutics, especially reabsorption of cholesterol and absorption of lipids in the intestine. Although liver can synthesize and secrete VLDL into circulation, the main source is food and excessive adipose tissue. Also, could be relevant for hepatic lipid storage disorders (not just cardiovascular).

PLA2G12B is categorized as secreted phospholipase (as opposed to cytoplasmic). Since the authors have PLA2G12B antibody, it would be very interesting to see if PLA2G12B is present in serum/circulating lipoprotein particles in WT mice. As the authors mentioned, one possibility is restructuring membranes. Although PLA2G12B is catalytically inactive phospholipase, it is still categorized as phospholipase: it could be part of the secreted lipoprotein particle, embedded into the phospholipid layer together with other

lipoproteins (APOB, etc), with its function related to phospholipids rather than triglycerides or cholesterol (p3 line 115 - “ensure efficient transfer of TG to secretible nascent TRLs”). The authors show data consistent with such hypothesis. PLA2G12B mutants are able to secrete lipoprotein particles but they are small/dense. The accumulating micelles are small, highly curved, which is usually achieved by presence of certain phospholipids. Larger lipoprotein particles have less curvature, which might allow for proper stabilization/orientation of proteins necessary for lipidation of the particle.

Another reviewer also alluded to the fact that PLA2G12B is a secreted protein (See Reviewer 1 comment 3 above: “PLA2G12B is a secreted protein. How can it remain in the ER to fulfil all these actions?”).

Indeed, PLA2G12B is a member of the secreted family of phospholipases. However, one of the key conclusions of our paper is that “*although PLA2G12B is a deeply conserved member of the secreted phospholipase gene family, it does not act as a phospholipase, nor is it secreted*”. We have provided additional data and references to support that PLA2G12B is retained within the ER by tagging PLA2G12B with mScarlet and visualizing its distribution in cells and live animals (Fig. 3A,B), and showing that it is retained in the ER and undetectable in the cell culture media / soluble fractions of human cells (Fig. S4E,F, S9).

Lastly, it would be very interesting to see if these mutants are able to absorb dietary fats in the intestine into chylomicrons or if most lipids are simply excreted with the feces and, therefore, liver uses lipid accumulations to increase bile acids production in attempt to increase uptake in the intestine.

This is a very interesting point, as at this point the fate of dietary fats absorbed by the intestine remains unclear.

Reviewer 4:

In the manuscript entitled “Pla2g12b Drives Expansion of Triglyceride-Rich Lipoproteins” Thierer and colleagues investigate the process of loading triglycerides onto lipoproteins for transport in serum by examining the optical properties of the yolk syncytial layer of larval zebrafish as an indicator of defects in lipid packaging. They identify an existing zebrafish mutant sa659, with a splice defect in gene pla2g12b, that is characterized by a darkened yolk phenotype. Using comprehensive phenotyping and elegant transgenic rescue approaches in combination with in vitro studies, the cellular localization at the ER membrane and functional protein domains are identified. This process is dependent on calcium binding by PLA2G12B. The importance of PLA2G12B function is further demonstrated in a murine model of atherosclerosis, where Pla2g12b mutant mice are hypolipidemic and protected from cardiac disease. Further, prior GWAS reveal polymorphisms of PLA2G12B promoter linked to plasma triglyceride levels.

Overall, this is a very exciting and comprehensive study that enables new insight into the transport and packaging of triglycerides into lipoproteins. It is of direct clinical and therapeutic relevance as it identifies a novel potential target to modulate serum lipid

levels. The study is extremely elegant, starting from a zebrafish mutant, extensive functional characterization, and involving human in vitro and genomic approaches and elegant murine long-term studies. The figures illustrate the enormous breadth of the studies undertaken, and the illustrations help the reader to understand the functional concepts. There are no major criticisms.

We would like to thank this reviewer for their thoughtful evaluation of our work.

Reviewer #1 (Remarks to the Author):

The authors have revised the text, and responded to the comments.

1. The authors state that PLA2G12B is not secreted. However, there are multiple reports stating the opposite. For example, <https://www.proteinatlas.org/ENSG00000138308-PLA2G12B>
<https://www.ncbi.nlm.nih.gov/gene/84647>
PMID: 21274867

2. Reviewer 1 comment 5: This is a critical question, and the authors reply: "We therefore speculate that differences the stage of development, rate of TRL production, and levels of luminal lipase expression (such as Triacylglycerol hydrolase, TGH/Ces3)^{18,68} may drive differences in CLD and LLD lipid partitioning between the YSL and adult tissues". Can this be tested?

Reviewer #2 (Remarks to the Author):

The authors have addressed questions and the paper has been improved.

Page 8 line 390 "...suggested that PLA2G12B mutant animals were remarkably healthy despite evidence of steatosis (Fig.S11)." It should be Fig.S12 in the revised manuscript.

Reviewer #3 (Remarks to the Author):

Dear Authors,

Thank you for addressing the questions raised in my initial review. The revised manuscript is much improved. Just a few very minor editorial comments: 1) introduction, 2nd paragraph, line 43 -TRLs is mentioned in main text for the first time (i.e. other than abstract) -> Triglyceride Rich Lipoprotein (TRLs); 2) page 2, line 98 Microsomal Triglyceride Transfer Protein -> MPT was already abbreviated in line 36; 3) lines 194-195 -> Pla2g12b - gene name ->italicize? Also, italicize on lines 637-646?

Point-by-point response to reviewers:

Reviewer 1:

Reviewer 1 comment 1:

The authors state that PLA2G12B is not secreted. However, there are multiple reports stating the opposite. For example, <https://www.proteinatlas.org/ENSG00000138308-PLA2G12B> <https://www.ncbi.nlm.nih.gov/gene/84647> PMID: 21274867

We have also encountered numerous sources stating that PLA2G12B is secreted. However, closer inspection of the underlying data revealed that experimental support for these claims is weak or non-existent. For example:

Link 1: The protein atlas (link 1 above, <https://www.proteinatlas.org/ENSG00000138308-PLA2G12B>) has classified PLA2G12B as secreted to blood. We evaluated the experimental evidence supporting this claim, and found that PLA2G12B was detectable in the human serum via mass-spectrometry, but curiously was **not detectable in serum by immunoassay or by proximity extension assay** (see screen capture below).

PROTEINS IN BLOOD	
Secretome annotation ⁱ	Secreted to blood
Detected in blood by immunoassay ⁱ	No
Detected in blood by mass spectrometry ⁱ	Yes
Detected in blood by proximity extension assay ⁱ	No

We then investigated the underlying mass-spectrometry data more closely (<https://www.proteinatlas.org/humanproteome/blood+protein/proteins+detected+in+ms>). These data are prefaced with the statement that “*This data set contains estimated concentrations for 4066 proteins, including both proteins that are actively secreted to plasma, but also locally secreted proteins and intracellular proteins that leak into the blood stream*”. We see that the first ~200 most abundant proteins identified are highly enriched in proteins that are genuinely secreted to the bloodstream (see red bars in screen capture below), and that proteins ranking between ~200-400 in serum abundance are enriched in “other” secreted proteins (green bars), but that proteins with rankings >800 are predominantly membrane proteins (blue bars) or intracellular proteins that leak into the bloodstream (purple bars). PLA2G12B was the 1,493rd most abundant protein detected in the serum, with a predicted concentration of 6.3 uq/L.

PLA2G12B is therefore less abundant in the bloodstream than many known intracellular proteins (such as TNNI3, troponin I3, shown in purple above). It is therefore clear that the detection of proteins in the human bloodstream via mass spectrometry is not sufficient to classify a protein as secreted.

This begs the question as to why PLA2G12B was annotated as a “secreted to blood” protein when many proteins with similar abundance are intracellular. We suspect that the presence of a signal peptide, as well as its close evolutionary relationship with secreted phospholipase genes led this protein to be annotated as secreted. We intend to initiate an inquiry with this database to encourage them to evaluate their own evidence in conjunction with our data to update the annotation.

Link 2: The NCBI link (Link 2 above, <https://www.ncbi.nlm.nih.gov/gene/84647>) also states that PLA2G12B is secreted. However, the only evidence supporting this claim falls under evidence code IEA (see screenshot below).

Component	Evidence Code	Pubs
located_in_extracellular_region	IEA	

According to the evidence code guideline found here (<https://geneontology.org/docs/guide-go-evidence-codes/>), IEA denotes that the evidence is “Inferred from Electronic Annotation”. The guideline further describes that “IEA-supported annotations are ultimately based on either homology and/or other experimental or sequence information, but **cannot generally be traced to an experimental source**”.

Link 3: The link to an original publication (link 3 above, <https://pubmed.ncbi.nlm.nih.gov/21274867/>) also states that PLA2G12B is secreted, but provides no evidence to support this claim. In the abstract, the authors introduce PLA2G12B as “Secreted phospholipase A₂ GXIIB (PLA₂GXIIB)” (see excerpt below):

“Hepatocyte nuclear factor-4 alpha (HNF-4a) is an important transcription factor governing the expression of genes involved in multiple metabolic pathways. Secreted phospholipase A₂ GXIIB (PLA₂GXIIB) is an atypical member of a class of secreted phospholipases A₂. We establish in this study that PLA₂GXIIB is an HNF-4a target gene. “

In the introduction, it becomes clear that this nomenclature is based off its evolutionary relationship with other “secreted PLA₂s (sPLA₂s)”.

In the conclusion, the authors again allude to PLA2G12B being secreted: “*Because PLA₂GXIIB is a secreted protein, its action may extend to tissues other than the liver to affect the homeostasis of fatty acids*”, but notably no reference is provided to support this conclusion.

This publication **provides no experimental evidence to suggest that PLA2G12B is secreted, nor does it reference other publications containing such evidence.**

Overall conclusion: In summary, we find that the classification of PLA2G12B as a secreted protein hinges on two observations: (1) that it is evolutionarily related to secreted phospholipase genes, and (2) that it is detectable in the bloodstream via mass-spectrometry. While inferring the localization of a protein based on sequence homology may be a useful starting point for generating hypotheses, it is not a valid substitute for experimental evidence. While PLA2G12B is detectable in the bloodstream via mass-spectrometry, it is detectable at levels comparable to canonical intracellular proteins. By their own omission, curators of this mass spectrometry dataset acknowledge that many of the proteins they detect are intracellular proteins that leak into the bloodstream. We conclude that the previous annotations of PLA2G12B as a secreted protein represent reasonable inferences, but are not supported by any experimental evidence. As the unexpected intracellular localization of PLA2G12B is a central finding of the paper, we have added a section to the discussion that specifically addresses these points:

“Although Pla2g12b was previously thought to be a secreted protein, this annotation lacks experimental support and appears to be based almost exclusively on the close evolutionary relationship between pla2g12b and other secreted phospholipase A₂ genes³⁶. We provide experimental evidence supporting that PLA2G12B is actively retained within the ER in all model systems tested, and is not detectable in the serum or media. Although

trace amounts of PLA2G12B have been detected in the human bloodstream using mass spectrometry⁵⁰, this sensitive technique routinely identifies canonical intracellular proteins that leak into the bloodstream. We therefore provide the first experimental evidence regarding the localization of Pla2g12b, and find that it is not actively secreted, and instead localizes within the ER."

Reviewer 1 comment 2:

Reviewer 1 comment 5: This is a critical question, and the authors reply: "We therefore speculate that differences the stage of development, rate of TRL production, and levels of luminal lipase expression (such as Triacylglycerol hydrolase, TGH/Ces3)18,68 may drive differences in CLD and LLD lipid partitioning between the YSL and adult tissues"." Can this be tested?

We agree that this is a very interesting question, but it is outside of the scope of the current study. This study is focused on elucidating the function of PLA2G12B, and we find that it is essential for TRL expansion and identify several functional domains contributing to that activity, and catalog broader changes in physiology in response to disruption of PLA2G12B. The observation that luminal lipid droplets appear in the YSL of PLA2G12B mutant zebrafish larvae demonstrates that PLA2G12B is not essential for the transfer of lipids into the lumen of the ER, allowing us to situate PLA2G12B downstream of MTP in the TRL assembly pathway using epistasis analyses (Fig. S2D). The observation that lipids accumulate in cytoplasmic lipid droplets in adult tissues is an expected (and previously published) consequence of lipid retention in TRL-producing tissues. Thus, understanding the partitioning of excess lipids that are not incorporated into TRLs between luminal and cytoplasmic lipid droplets is of high academic interest and a future focus of our research program, but is unlikely to provide additional insight into the role of PLA2G12B in TRL assembly. We have added the following sentence to this section of the discussion to more clearly emphasize the key conclusion of our lipid droplet analyses:

"However, the observation of LLDs in the YSL of pla2g12b mutant zebrafish demonstrates that Pla2g12b is not essential for the transfer of lipids into the ER lumen."

Reviewer 2:

Reviewer 2 comment 1:

Page 8 line 390 "...suggested that PLA2G12B mutant animals were remarkably healthy despite evidence of steatosis (Fig.S11)." It should be Fig.S12 in the revised manuscript.

This has been corrected.

Reviewer 3:

Reviewer 3 comment 1:

Introduction, 2nd paragraph, line 43 –TRLs is mentioned in main text for the first time (i.e. other than abstract) → Triglyceride Rich Lipoprotein (TRLs)

This has been corrected.

Reviewer 3 comment 2:

Page 2, line 98 Microsomal Triglyceride Transfer Protein -> MPT was already abbreviated in line 36

This has been corrected.

Reviewer 3 comment 3:

Lines 194-195 -> Pla2g12b - gene name ->italicize? Also, italicize on lines 637-646?

While we agree that this style often interferes with readability, we have done our best to adhere to the gene and protein nomenclature guidelines for various species throughout the manuscript. For zebrafish, the guidelines dictate that the gene name is all lower-case and italicized (*pla2g12b*), whereas the corresponding protein product is not italic and has the first letter capitalized (Pla2g12b, see guidelines available here: <https://zfin.atlassian.net/wiki/spaces/general/pages/1818394635/ZFIN+Zebrafish+Nomenclature+Conventions>). This means that the capitalization and italicization scheme may differ even within the same paragraph (see excerpt below), but is meant to distinguish between the coding gene and its protein product. The guidelines for mouse genes and proteins (*Pla2g12b* and PLA2G12B) and human genes and proteins (*PLA2G12B* and PLA2G12B) also differ in their capitalization schemes. Please let us know if there are instances where we have departed from these guidelines and they will be amended.

“...poorly characterized gene *pla2g12b*, which encodes a catalytically inactive phospholipase enzyme with no known function³⁵. Although previous studies have suggested that Pla2g12b plays a role in lipid or lipoprotein metabolism^{11,34,36-39} ...”

Reviewer #1 (Remarks to the Author):

Revisions OK

Reviewer #2 (Remarks to the Author):

The authors have addressed all my questions. I recommend the manuscript for publication.

Reviewer #3 (Remarks to the Author):

The authors has addressed my comments.